# Structural basis for ion selectivity in TMEM175 K$^+$ channels

Janine D Brunner[1,2,3,4,5]*, Roman P Jakob[2†], Tobias Schulze[6†], Yvonne Neldner[1‡], Anna Moroni[7], Gerhard Thiel[6], Timm Maier[2], Stephan Schenck[1,3,4,5]*

[1]Department of Biochemistry, University of Zürich, Zürich, Switzerland; [2]Department Biozentrum, University of Basel, Basel, Switzerland; [3]Laboratory of Biomolecular Research, Paul Scherrer Institut, Villigen, Switzerland; [4]VIB-VUB Center for Structural Biology, VIB, Brussels, Belgium; [5]Structural Biology Brussels, Vrije Universiteit Brussel, Brussels, Belgium; [6]Membrane Biophysics, Technical University of Darmstadt, Darmstadt, Germany; [7]Department of Biosciences, University of Milano, Milan, Italy

*For correspondence:
janine.brunner@vub.be (JDB);
stephan.schenck@vub.be (SS)

[†]These authors contributed equally to this work

Present address: [‡]Department of Trauma, University Hospital Zürich, Zürich, Switzerland

Competing interests: The authors declare that no competing interests exist.

**Abstract** The TMEM175 family constitutes recently discovered K$^+$ channels that are important for autophagosome turnover and lysosomal pH regulation and are associated with the early onset of Parkinson Disease. TMEM175 channels lack a P-loop selectivity filter, a hallmark of all known K$^+$ channels, raising the question how selectivity is achieved. Here, we report the X-ray structure of a closed bacterial TMEM175 channel in complex with a nanobody fusion-protein disclosing bound K$^+$ ions. Our analysis revealed that a highly conserved layer of threonine residues in the pore conveys a basal K$^+$ selectivity. An additional layer comprising two serines in human TMEM175 increases selectivity further and renders this channel sensitive to 4-aminopyridine and Zn$^{2+}$. Our findings suggest that large hydrophobic side chains occlude the pore, forming a physical gate, and that channel opening by iris-like motions simultaneously relocates the gate and exposes the otherwise concealed selectivity filter to the pore lumen.

## Introduction

Potassium is the major intracellular cation and crucial to many fundamental cellular processes such as maintenance of the resting membrane potential, repolarization of action potentials, counter-ion flux and osmoregulation. The function and distribution of K$^+$ ions in endomembrane compartments such as endosomes and lysosomes are less clear and their significance is only recently beginning to emerge (*Feng et al., 2018*). Lysosomes, the recycling organelles of the cell, are characterized by a very low luminal pH for the efficient decomposition of its contents. They have been recognized lately as a central hub in metabolic regulation of the cell (*Perera and Zoncu, 2016*; *Lamming and Bar-Peled, 2019*). Like in other endomembrane compartments, ion channels and transporters, in conjunction with the vacuolar-type ATPase, are essential for the regulation of the luminal pH, the membrane potential and organelle fusion and thus regulate the transport of other solutes across the membrane and also the dynamics and fate of these organelles (*Li et al., 2019*). It has been known for decades that lysosomal membranes are permeable for K$^+$ and even more so for Cs$^+$, a hallmark of this type of endo-membranes (*Henning, 1975*). The underlying channel has recently been identified as a member of the transmembrane protein family 175 (TMEM175). The TMEM175 channel was found to mediate a major K$^+$ permeability of lysosomes and late endosomes (hence also named K$_{EL}$) (*Cang et al., 2015*), and is not related to canonical K$^+$ channels. TMEM175 channels are present in animals, eubacteria and archaea but are not found in plants and fungi. They exhibit selectivities ranging from P$_K$/P$_{Na}$ of ~35-20 (*Cang et al., 2015*; *Lee et al., 2017*) in vertebrates to P$_K$/P$_{Na}$ of ~2–5 in bacteria and have been described as 'leak-like' channels (*Cang et al., 2015*). TMEM175 channels,

unlike canonical $K^+$ channels, conduct $Cs^+$ ions and are not blocked by $Ba^{2+}$, tetraethylammonium or quinine but instead by $Zn^{2+}$ ions. Like other $K^+$ channels, they are blocked by 4-aminopyridine (with the exception of bacterial TMEM175 proteins), and conduct $Rb^+$ but not $Ca^{2+}$ and N-methyl-D-glucamine (*Cang et al., 2015*). The vertebrate TMEM175 proteins are composed of two homologous non-identical repeats, each comprising six transmembrane domains forming dimers; the bacterial homologues consist of only one such repeat and form tetramers (*Cang et al., 2015*; *Lee et al., 2017*). In prokaryotes, the function of TMEM175 proteins is currently unclear but may be linked to the regulation of the membrane potential (*Cang et al., 2015*). In vertebrates, deletion of the TMEM175 gene leads to increased lysosomal pH under conditions of starvation, reduced proteolytic activity in lysosomes and aberrant autophagosome fusion and clearance (*Cang et al., 2015*; *Jinn et al., 2017*). The human TMEM175 channel has been linked to Parkinson disease (PD) by several genome wide association studies and is considered as a highly significant risk gene for the early onset of this neurodegenerative disease (*Jinn et al., 2017*; *Nalls et al., 2014*; *Chang et al., 2017*). The deficiency in autophagosome clearance and the impaired proteolytic activity by loss of TMEM175 are presumably causative for the accumulation of insoluble α-synuclein fibrils in PD models and could explain why this channel is relevant for the progression of PD (*Jinn et al., 2017*). In particular with its channel function, TMEM175 may thus be a viable drug-target to interfere with the pathogenesis.

Selectivity for $K^+$ ions, with the exception of the very weakly selective trimeric intracellular cation (TRIC) channels (*Su et al., 2017*), is intimately associated with a P-loop architecture (*Doyle et al., 1998*). The lack of such a P-loop motif in TMEM175 channels hence raises the question how $K^+$ conduction and selectivity is achieved in this unrelated architecture. To gain insight into the structure and mechanisms of this new family of ion channels we structurally characterized a bacterial TMEM175 member. We obtained a crystal structure at a resolution of 2.4 Å of a TMEM175 channel from *Marivirga tractuosa* (MtTMEM175) and in combination with functional analysis, propose how selectivity for $K^+$ ions is achieved. We further provide data that explains the increased potassium selectivity as well as the pronounced sensitivity towards 4-aminopyridine and zinc of human TMEM175 ($K_{EL}$) compared to the bacterial counterparts and suggest events that lead to channel opening. Importantly, our data and conclusions deviate from a previously reported structure and functional investigation (*Lee et al., 2017*), in particular regarding the principle of $K^+$ selectivity.

## Results

### Crystallization and general architecture of MtTMEM175

From an expression screen of over 30 bacterial TMEM175 channels we identified several homologues as candidates for a structural characterization. However, the crystals for all of the tested homologues, including MtTMEM175, were diffracting maximally to a resolution of 10 Å even in complex with nanobodies (Nbs). Finally, MtTMEM175 was crystallized in complex with a Nb that in turn was engineered by fusing an N-terminally truncated Maltose Binding Protein (MBP) to its C-terminus (*Figure 1a,b* and *Figure 1—figure supplement 1a,b*) which greatly improved diffraction. We solved the structure at a resolution of 2.4 Å using highly redundant data by molecular replacement based on this novel Nb-MBP chaperone, with structures of Nbs and maltose-bound MBP as search models (*Supplementary file 1*) and could build a map of high quality (*Figure 1—figure supplement 2*, *Figure 1—figure supplement 3* and *Figure 1—figure supplement 4*). We named this chaperone scaffold 'macrobody' (Mb) and termed the Mb used in this study $Mb_{51H01}$. Macrobodies could develop into a promising tool for structural biology applications, especially in electron cryo-microscopy, similar to the recently reported megabodies (*Laverty et al., 2019*). Each MtTMEM175 subunit is composed of six transmembrane helices (*Figure 1c*) which assemble to form a tetrameric channel as verified using SEC-MALLS of uncomplexed MtTMEM175 (*Figure 1—figure supplement 1c*). Helix one is the pore-lining helix, as predicted earlier (*Cang et al., 2015*), and constitutes the highest degree of conservation (*Figure 1c* and *Figure 1—figure supplement 1d*).

The MtTMEM175 structure reveals a network of hydrogen bonds in proximity to the intracellular pore entrance that positions helices 1–3 relative to each other and interconnects adjacent subunits (*Figure 1d,e*). The network is similar to the one of a recently reported structure of a TMEM175 homologue (CmTMEM175, PDB accession 5VRE) (*Lee et al., 2017*; *Figure 1—figure supplement*

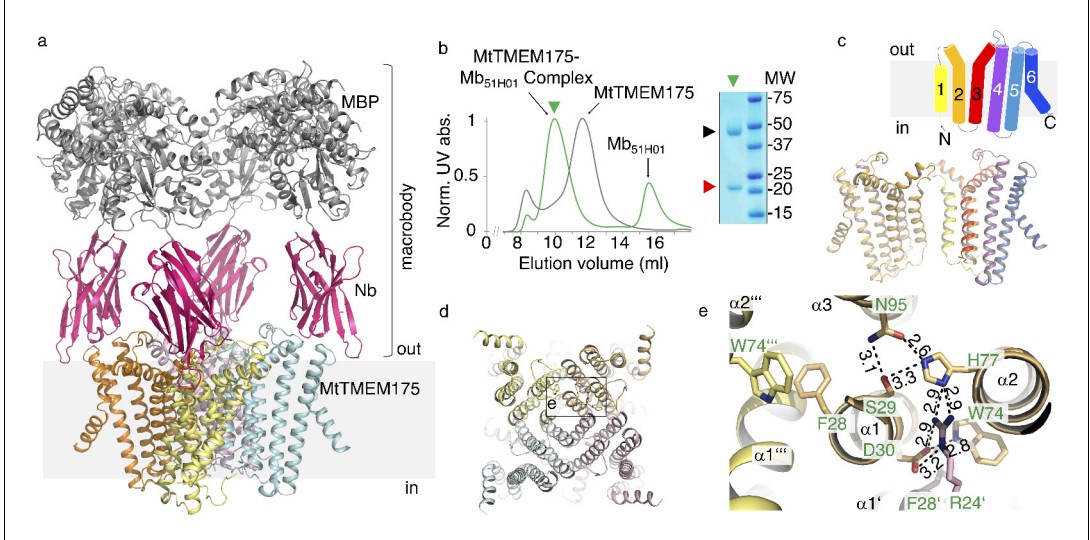

**Figure 1.** Structure of the MtTMEM175-Mb$_{51H01}$ complex. (**a**) Side view of the complex with MtTMEM175 channel and macrobody. Approximate membrane boundaries are indicated. (**b**) Left: Size exclusion chromatogram of MtTMEM175 (grey) and MtTMEM175-Mb$_{51H01}$ complex (green). Right: Coomassie-stained SDS-PAGE gel of the peak fraction (green triangle) indicating complex formation of MtTMEM175 (red triangle) with Mb$_{51H01}$ (black triangle). (**c**) Arrangement of transmembrane helices 1–6 in MtTMEM175. N- and C-termini are indicated. Two subunits are omitted for clarity. (**d**, **e**) MtTMEM175 tetramer (**d**) and close-up view on interacting residues with corresponding numbering (**e**). Distances are given in Å. The view is from the intracellular side.

The online version of this article includes the following source data and figure supplement(s) for figure 1:

**Figure supplement 1.** Macrobody, sequence alignment of TMEM175 proteins and tetramer assembly.

**Figure supplement 2.** Electron density map of MtTMEM175 transmembrane helices.

**Figure supplement 3.** Electron density map of MtTMEM175.

**Figure supplement 4.** Plots of I/sigI vs. resolution.

**Figure supplement 4—source data 1.** Raw data for *Figure 1—figure supplement 4*.

*1e,f*) and involves most of the highly conserved residues in TMEM175 channels, also the FSD motif (Phe28, Ser29, Asp30 in MtTMEM175), three consecutive amino acids at the N-terminus of helix one that were originally proposed to play a role in ion conduction (*Figure 1—figure supplement 1d*; *Cang et al., 2015*). Generally, the level of conservation for TMEM175 channels is strikingly low in transmembrane helices 4–6, persuading us to exclude this region largely from our analysis. The interactions of Ser29, His77 and Asn95 are present in both structures and position the first three transmembrane helices relative to each other. Trp74 from helix two is interacting with Asp30 in helix 1 of the same subunit in MtTMEM175, whereas in CmTMEM175 this tryptophan is more involved in a cation-π stack with the phenylalanine from the FSD motif of the adjacent subunit (*Figure 1e* and *Figure 1—figure supplement 1e,f*). Different from CmTMEM175, Arg24 (another highly conserved residue in TMEM175 channels) is interacting with His77 and Asp30 of the adjacent subunit, thereby connecting neighboring subunits (*Figure 1—figure supplement 1g*). Gel filtration profiles of Arg24 mutant proteins support a role in tetramer assembly (*Figure 1—figure supplement 1h*). We can thus assign at least one role for most of the highly conserved residues in MtTMEM175, which makes us confident that MtTMEM175 provides a clear case to characterize the TMEM175 family in general. Importantly, the conservation of these key residues in human TMEM175 suggests an overall very similar architecture. None of the aforementioned residues seems to play a distinct role in selectivity that would be immediately apparent from the structure.

## MtTMEM175 is a weakly selective K$^+$ channel

For electrophysiological characterization MtTMEM175 was expressed in HEK293 cells as previously done with homologues from *Streptomyces collinus* and *Chryseobacterium sp.* (ScTMEM175 and CbTMEM175) (*Cang et al., 2015*). In whole cell patch clamp experiments we recorded only from transfected cells non-rectifying, non-inactivating K$^+$ currents that showed no signs of voltage-

dependence (*Figure 2a*). These currents were blocked by $Zn^{2+}$ ions and at the applied concentration of 5 mM also in a voltage-independent manner (*Figure 2b*). Similar to ScTMEM175 and CbTMEM175, MtTMEM175 has also a low selectivity for $K^+$ ($P_K/P_{Na} \sim 4.3$). It conducts $Cs^+$ and $Rb^+$ with a similar efficiency as $K^+$, and to lesser extent, similar to $Na^+$, also $Li^+$ (*Figure 2c* and *Figure 2— figure supplement 1*). The channel has no apparent permeability for anions; the reversal voltage was not significantly different when the same recordings were done with standard bath solution

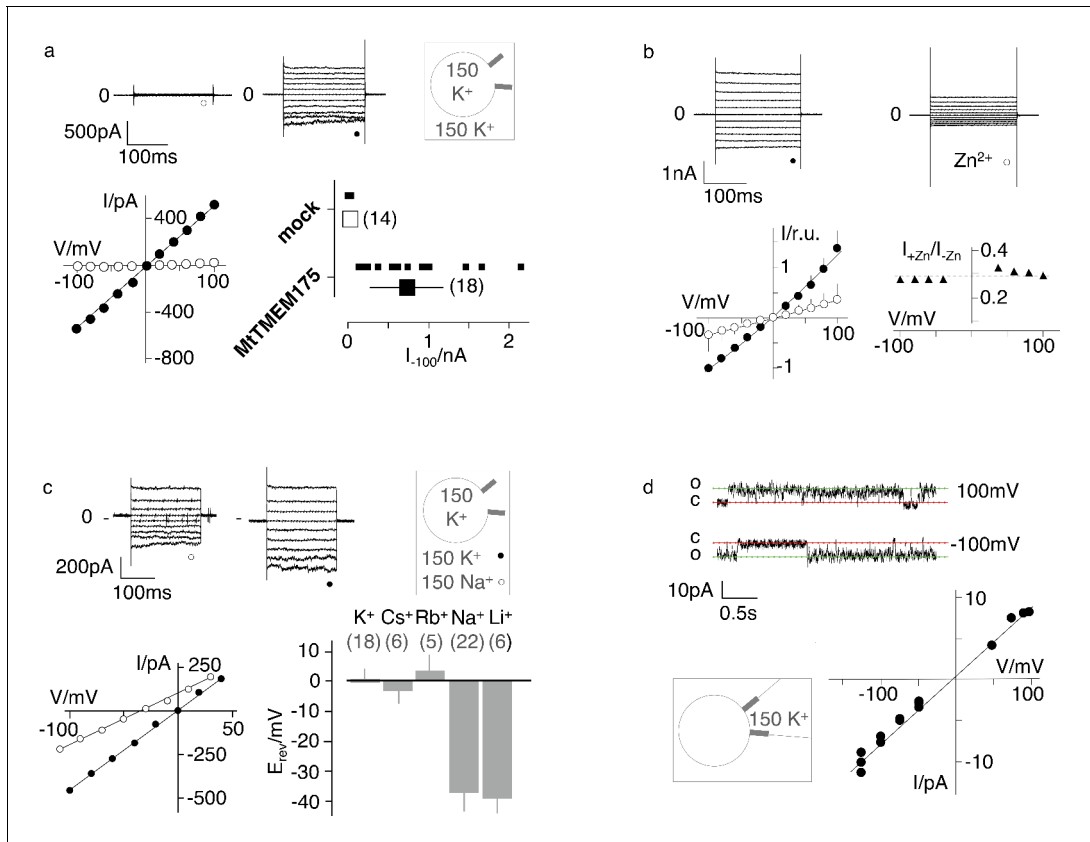

**Figure 2.** Electrophysiological characterization of MtTMEM175. (a) Current responses to standard voltage pulse protocol in mock (○) and MtTMEM175 (●) transfected HEK293 cells (upper panel) and corresponding steady state I/V relations (lower left). Plot of currents recorded in same manner at −100 mV for individual cells (small symbols) and mean ±s.d. (large symbols) (lower right). Number of cells in brackets. (b) HEK293 cells expressing MtTMEM175 before (●) and after (○) adding 5 mM $ZnSO_4$ to the bath solution containing 150 mM $K^+$ (upper panel). Mean I/V relation (bottom left) of n = 4 cells (±s.d.). To compare the effect on different cells the I/V relation was normalized to currents at −100 mV in the absence of blocker (bottom left). The voltage dependency of the $Zn^{2+}$ block was estimated by dividing currents in the presence and absence of $Zn^{2+}$ (I+Zn/I-Zn) (bottom right) (c) HEK293 cells expressing MtTMEM175 (top row) before (left) and after (middle) replacing $Na^+$ (○) with $K^+$ (●) in the external buffer and corresponding I/V relation (bottom left). Same experiments were performed by exchanging $K^+$ in external buffer by other cations. The mean reversal voltage ($E_{rev}$) (±s.d., number of cells in brackets) is depicted in lower right panel. (d) Exemplary channel fluctuations at ±100 mV measured in cell-attached configuration on HEK293 cells expressing MtTMEM175 (upper) and pooled unitary I/V relation of single channel events from 10 measurements in four different cells (lower right) using standard bath and pipette solutions.

The online version of this article includes the following source data and figure supplement(s) for figure 2:

**Source data 1.** Raw data for *Figure 2a*.
**Source data 2.** Raw data for *Figure 2b*.
**Source data 3.** Raw data for *Figure 2c*.
**Source data 4.** Raw data for *Figure 2d*.
**Figure supplement 1.** $K^+$ selectivity in ScTMEM175 and electron density of a putative maltoside in the MtTMEM175 structure.
**Figure supplement 1—source data 1.** Raw data for *Figure 2—figure supplement 1*.
**Figure supplement 2.** Plasma membrane localization of TMEM175 proteins.
**Figure supplement 3.** Cell surface labelling of MtTMEM175 using Nb$_{51H01}$-vYFP.
**Figure supplement 4.** Purification of MtTMEM175 from HEK cells.
**Figure supplement 4—source data 1.** Raw data for *Figure 2—figure supplement 4*.

containing the large anion methanesulfonate (+0.64 ± 3 mV n = 18) or in a bath with 150 mM KCl (2.4 ± 4, n = 7). We obtained a few single channel recordings from MtTMEM175-transfected cells that revealed a unit conductance of ~70 pS and showed channel flickering (*Figure 2d*). We do not have definitive proof that these currents originate from MtTMEM175, however several arguments support this view. First, we recorded them only in transfected cells, which exhibited typical MtTMEM175 macroscopic currents after breaking into the whole cell configuration. Second, like the macroscopic MtTMEM175 current also the single channel I/V relation reverses around 0 mV (*Figure 2d*). Finally, the unitary conductance of the channels at −100 mV is in the range of the conductance of mock transfected HEK293 cells in whole cell mode (*Figure 2a*), making it unlikely that the currents originate from endogenous channels. The presence of gating events is inconsistent with the definition of a leak channel and has significance for the interpretation of the structure.

Albeit currents have been recorded of HEK cells transfected with CbTMEM175, ScTMEM175 and MtTMEM175 ( [*Cang et al., 2015*] and this study) functional expression of these bacterial channels at the plasma membrane has not been confirmed by other methods. Overexpressed MtTMEM175 with a C-terminal Venus-YFP (vYFP) indicated a wide distribution, without a prominent presence at the plasma membrane (*Figure 2—figure supplement 2a*). Thus, we generated plasma membrane patches by decapitation of cells with ice cold distilled water (*Biel et al., 2016*). In the membrane from transfected and control cells we could show with TIRF microscopy that vYFP-tagged MtTMEM175 and hTMEM175 also co-localize with specific plasma membrane stains in HEK293 cells (*Figure 2—figure supplement 2b*). In a second experiment we purified C-terminally vYFP-tagged $Nb_{51H01}$ ($Nb_{51H01}$-vYFP) from transiently transfected HEK cells (*Figure 2—figure supplement 3a*) and incubated unfixed HEK cells that were mock-transfected or transfected with non-fluorescent MtTMEM175 with the purified fluorescent $Nb_{51H01}$-vYFP. $Nb_{51H01}$ recognizes an extracellular epitope of MtTMEM175 (*Figure 1a*) and is thus suitable to label non-permeabilized cells that expose MtTMEM175 on the plasma membrane. *Figure 2—figure supplement 3* clearly shows that $Nb_{51H01}$-vYFP is only labeling HEK cells transfected with MtTMEM175 and provides evidence for at least partial plasma membrane localization and correctly folded MtTMEM175 channels. Further, non-fluorescent MtTMEM175 purified from transiently transfected HEK cells elutes as a tetramer in SEC at near identical volumes as bacterially expressed MtTMEM175 (*Figure 2—figure supplement 4*). Together, these results are in support of functional plasma membrane expression of MtTMEM175 in HEK cells and in concordance with our electrophysiology data.

## The MtTMEM175 structure reveals hydrated and dehydrated K+ ions

The structure of MtTMEM175 revealed two densities attributable to K+ ions, termed 1K+ and 2K+ (*Figure 3a–c*). The presence of K+ is supported by data collection at higher wavelengths of 2.02460 Å (*Figure 3—figure supplement 1a,b* and *Supplementary file 1*). In contrast, the structure of CmTMEM175, which was solved at 3.3 Å did not reveal bound ions, even in crystals soaked with heavier monovalent and divalent ions (*Lee et al., 2017*). One K+ ion (1K+, occupancy ~1) in the MtTMEM175 structure, is located at an ion binding site at the extracellular pore entrance (*Figure 3b*). This binding site resembles a short selectivity filter (*Chen et al., 2017*; *Guo et al., 2017*; *Shen et al., 2016*). The K+ ion is complexed by eight water molecules in an anti-prismatic geometry (*Figure 3—figure supplement 1c*) that are coordinated by backbone oxygens of Leu42, Ser43 and Ser44 (*Figure 3b,d* and *Figure 3—figure supplement 1d*). The respective backbone oxygens of these residues are 12, 13.1 and 14.9 Å apart (*Figure 3d*). Except for the conserved Leu42, no obvious motif for this region is apparent (*Figure 1—figure supplement 1d*).

The second K+ ion (2K+, occupancy ~0.5) is not coordinated by water molecules which suggests the permeation of dehydrated K+ ions in these channels. The K+ ion at position 2K+ is located between the layers of Leu35 and Thr27 (*Figure 3a,c*). It is likely that this K+ ion is trapped in the pore due to the restriction at Thr27 and the lack of clear interactions with the protein. It is further worth to note that the density at 2 K+ is likely also partly constituted by Na+ ions. The anomalous signal (*Figure 3—figure supplement 1a,b*) at this position is proportionally weaker in comparison to the signal at 1K+ than in the native data set (*Figure 3b,c*). We can therefore only estimate that the occupancy at 2K+ is approximately ~0.5. By soaking crystals with Cs+ and Rb+ we detected clear anomalous density for both ions at the position of 1K+ (*Figure 3e*) providing additional evidence for an affinity towards monovalent cations with similar properties as K+ at this extracellular ion-binding site. No significant anomalous signal for Cs+ or Rb+ was found at the 2K+ position, indicating that

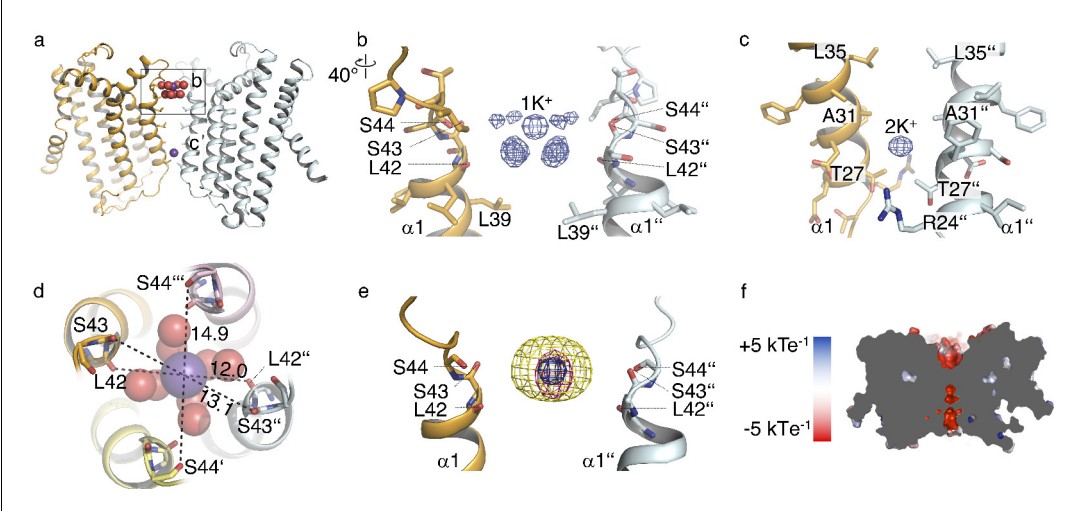

**Figure 3.** Ions in the MtTMEM175 structure. (**a–c**) Side view on MtTMEM175 (**a**) and close-up views of the ion binding site with a hydrated $K^+$ ion at position $1K^+$ (**b**) and another $K^+$ ion within the pore at position $2K^+$(**c**). In (**a**), $K^+$ ions and water molecules are displayed as purple and red spheres, respectively. In (**b**) and (**c**) the $2F_o$-$F_c$ electron density is depicted as blue mesh at the position $1K^+$ and $2K^+$ (at 2.4 Å, contoured at 1.8 σ, sharpened with b=-25). Two subunits are omitted for clarity. (**d**) Top view of the ion binding site. Distances between opposing backbone oxygens of Leu42, Ser43 and Ser44 are indicated in Å. Side chains are omitted and the size of the spheres is reduced for clarity. (**e**) Substitution of $K^+$ in the ion binding site with $Cs^+$ and $Rb^+$. The $2F_o$-$F_c$ electron density (as in (**b**) and (**c**), blue mesh) marks the position of the $K^+$ ion. Anomalous difference electron densities of $Cs^+$ (at 3.8 Å, contoured at 7 σ) and $Rb^+$ (at 3.6 Å, contoured at 7 σ, blurred with b = 125) are shown in yellow and magenta, respectively. (**f**) Illustration of the surface electrostatic potential across the pore.

The online version of this article includes the following figure supplement(s) for figure 3:

**Figure supplement 1.** $K^+$ coordination in MtTMEM175, $3_{10}$-helix in CmTMEM175 and analysis of the ion binding site region in CmTMEM175.

the channel would have to open for diffusion of these generally permeable ions to this position. Collectively, these results advocate the existence of a conductive conformation with a wider pore that is different from the crystal structure. Additional density in the $2F_o$-$F_c$ map on the extracellular side in proximity to the ion binding-site was attributed to a maltose moiety from a detergent molecule. We tested for potential influence of maltose on the conductance by electrophysiology, but could not detect any effects (*Figure 2—figure supplement 1*).

The hydrated $K^+$ ion in the MtTMEM175 crystal is reminiscent of the one in the vestibule of a high-resolution structure of KcsA in close proximity to the selectivity filter entrance (*Zhou et al., 2001*). In comparison, the two planes in the $K^+$-hydrate in MtTMEM175 are skewed, due to interactions with the surrounding backbone oxygens (*Figure 3—figure supplement 1c–e*). The eightfold coordination of $K^+$ ions in square antiprism is also seen inside the canonical selectivity filter, where it is mediated by backbone oxygens (*Figure 3—figure supplement 1e*; *Doyle et al., 1998*; *Zhou et al., 2001*). MtTMEM175 crystallized in the presence of equimolar amounts of $Na^+$ and $K^+$ which apparently did not interfere with $K^+$ coordination at $1K^+$. The ion binding site thus recapitulates a number of central elements seen in $K^+$ coordination of ion channels, in particular the coordination geometry. However, the low conservation, its simplicity, the indirect coupling of the $K^+$ ion to the backbone oxygens and the single binding site make it questionable that this region determines selectivity. It could be that this binding site serves to attract monovalent ions with similar properties as $K^+$ and plays a role in the resolvation or desolvation of $K^+$ ions that pass through the pore. Apart from the ion binding site, the negative electrostatic potential in the pore lumen would be another property promoting cation permeation (*Figure 3f*).

## Bulky residues constrict the pore and form a physical gate in the closed state

In the structure of MtTMEM175 Leu35 is occluding the pore to such an extent that $K^+$ ions could not pass (*Figure 4*). This bulky residue is thus likely constituting a hydrophobic physical gate. From single channel recordings and the lack of exchangeability of $2K^+$ with $Cs^+$ or $Rb^+$ we have indications for

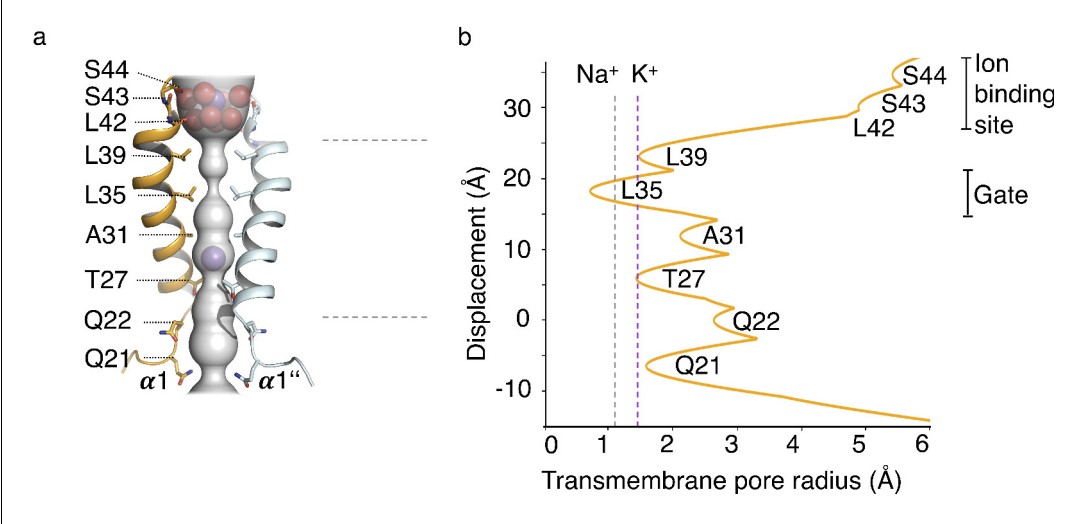

**Figure 4.** HOLE analysis. (**a**) The ion conduction pathway is illustrated as grey surface and pore-lining residues are displayed. K$^+$ ions and water molecules are shown as purple and red spheres, respectively. (**b**) The pore radius along the central axis is shown in Å. Dashed lines indicate the radii of K$^+$ and Na$^+$ ions without inner hydration shell.

The online version of this article includes the following figure supplement(s) for figure 4:

**Figure supplement 1.** Conservation in TMEM175 proteins.

open and closed conformations in support of a gate in TMEM175 channels. Due to the highly constricted pore, the structure of MtTMEM175 thus very likely represents a closed state and this implies that structural rearrangements have to take place in order for the channel to become conductive. Opening of the channel would inevitably require displacement of the hydrophobic side chain of Leu35 from the pore center, probably by a helix-rotation as seen in the NaK channel (*Alam and Jiang, 2009a*) or in TRPV6 (*McGoldrick et al., 2018*). Previously, the homologous residues in CmTMEM175 (Ile23) or human TMEM175 (Ile46 and Ile271) were described as the determinants for selectivity (*Lee et al., 2017*), however we propose that this position is generally occupied by a residue that acts as a physical gate for ions as discussed below in more detail. Furthermore, the pore-lining residues that physically interact with passing ions and determine conduction or selectivity would likely show the highest degree of conservation.

When we plotted the conservation of residues from an AL2CO analysis of randomly chosen TMEM175 proteins onto the structure of MtTMEM175 (see Materials and methods section) we found that the most highly conserved residues are not the bulky hydrophobic residues that face the pore in the observed conformation. The more conserved residues are located to the side of the pore-lining helix, facing helix 1 of the next subunit (*Figure 4—figure supplement 1*). In MtTMEM175, these are the residues Thr38, Ala34 and Asp30 (the latter being part of the FSD motif and involved in hydrogen bonds to Arg24 and Trp74). Threonine38 and Ala34 do not show any particular interaction with their respective environments, for example with the adjacent helix that they are facing, raising the question why these residues have such a high degree of conservation. In particular Thr38 is of interest since it is the most conserved residue among all TMEM175 proteins (*Figure 4—figure supplement 1e* and *Figure 5a,c*). Threonine38 forms a layer that is interspersed between Leu35 and Leu39 and participates in a bifurcated hydrogen bond with the main-chain oxygen of Ala34 (*Figure 5—figure supplement 1a,b*). A rotation of helix 1 (in clockwise direction when viewed from intracellular) as part of an iris-like opening of the gate (Leu35), would expose the side chain of Thr38 to the pore lumen. Following this line of thoughts, we reasoned that K$^+$ ions could interact with the side chain of Thr38 in a conductive conformation of MtTMEM175 and mutated Thr38 to alanine.

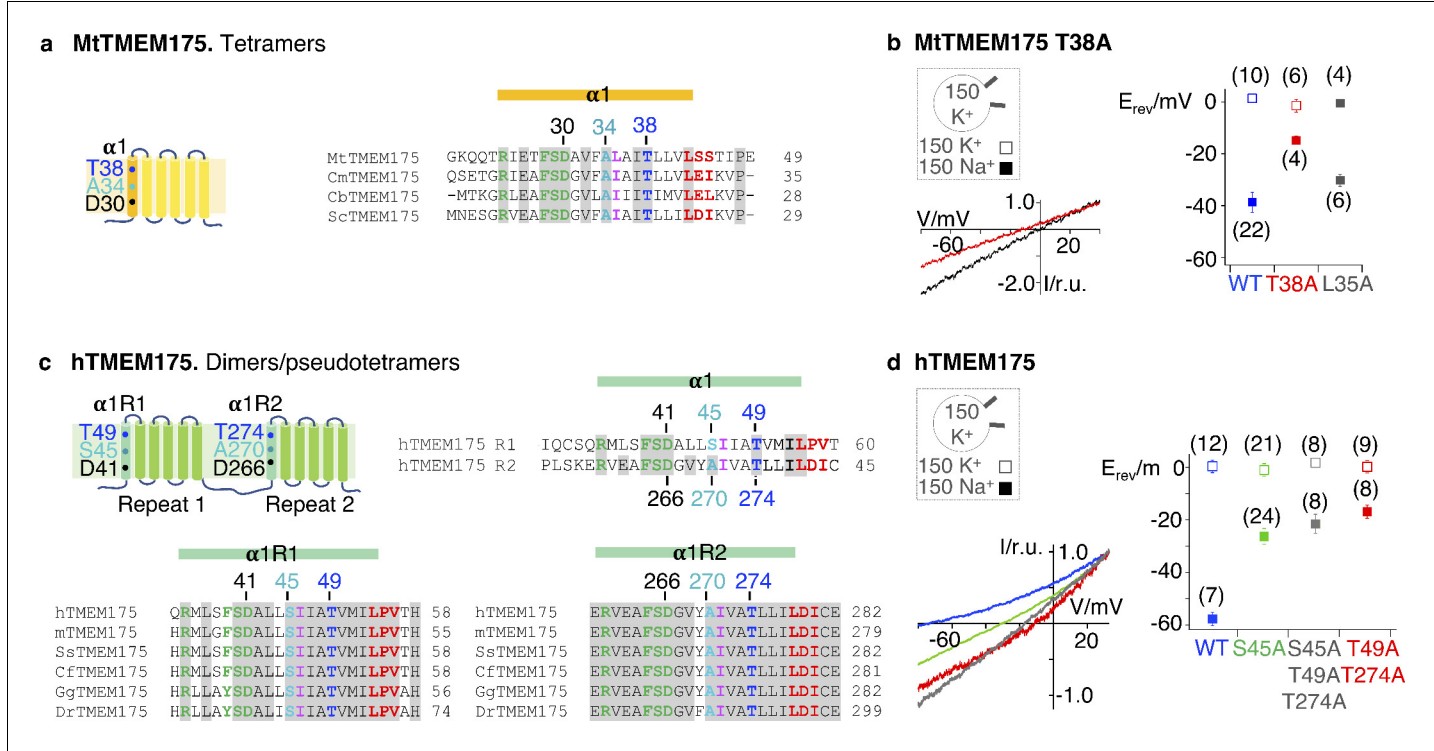

**Figure 5.** K⁺selectivity in TMEM175 proteins. (**a**) Subunit organization of MtTMEM175 and alignment of bacterial sequences highlighting the most conserved residues in helix 1. (**b**) Lower left: representative currents elicited by a ramp protocol (−80 to +40 mV in 200 ms) from HEK293 cells transfected with the MtTMEM175 T38A in external solution with 150 mM K⁺ (black) or Na⁺ (red); currents normalized to values at +33 mV. Right: plot of the average reversal potentials (E$_{rev}$ ±s.d.) for T38A or L35A mutants in comparison to WT in symmetrical buffer with 150 mM K⁺ (□) or in external buffer with 150 mM Na⁺ (■). Number of patched cells in brackets. (**c**) Subunit organization of hTMEM175 and alignment highlighting conserved residues. m: mouse, Ss: *Sus scrofa*, Cf: *Canis familiaris*, Gg: *Gallus gallus*, Dr: *Danio rerio* (**d**) Lower left: representative currents elicited by a ramp protocol as in (**a**) from HEK293 cells transfected with hTMEM175 WT (blue) or mutants S45A (green), T49A/T274A (red) or S45A/T49A/T274A (grey) in external solution with 150 mM Na⁺; currents normalized to values at +33 mV. Right: plot of the respective average reversal potentials (E$_{rev}$ ±s.d.) for each construct in symmetrical buffer with 150 mM K⁺ (□) or in external buffer with 150 mM Na⁺ (■). Number of patched cells in brackets.

The online version of this article includes the following source data and figure supplement(s) for figure 5:

**Source data 1.** Raw data for *Figure 5b*.
**Source data 2.** Raw data for *Figure 5d*.
**Figure supplement 1.** Threonine 38 in MtTMEM175, threonine and serine residues in the pore of MtTMEM175 and hTMEM175.

## The highly conserved Thr38 confers K⁺ selectivity to the MtTMEM175 channel

When analyzed in whole cell patch clamp recordings in HEK293 cells the T38A mutant of MtTMEM175 showed a strongly impaired selectivity for K⁺ ions, as exchanging K⁺ in the bath solution for Na⁺ caused only a minor shift of the reversal potential by −15 ± 2 mV (n = 4), corresponding to a P$_K$/P$_{Na}$ < 2 (*Figure 5b*). For comparison, the WT protein responds to a replacement of K⁺ for Na⁺ with a shift of −37 ± 6 mV (n = 22) (*Figure 2c* and *Figure 5b*). When we mutated Leu35 to alanine the channel showed only a slightly reduced selectivity compared to WT channels (E$_{rev}$ = −31 ± 2 mV, n = 6) (*Figure 5b*), in contrast to the findings on Ile23 for CmTMEM175 and Ile46/Ile271 in hTMEM175 (*Lee et al., 2017*). Our data speaks against a primary function in selectivity for these bulky hydrophobic residues as discussed below. We thus conclude that Thr38 plays a pivotal role for K⁺ selectivity and conductance in MtTMEM175, reflected also in its high degree of conservation (*Figure 4—figure supplement 1e*). Notably, the side chain of a conserved threonine is also essential for the coordination of K⁺ ions at the S4 position in the selectivity filter of canonical K⁺ channels. Hence, not only carbonyl ligands, but also the threonine side chain is suited to coordinate K⁺ ions with impact on selectivity and conductance (*Zhou and MacKinnon, 2003*; *Zhou and MacKinnon, 2004*; *Krishnan et al., 2008*; *Chatelain et al., 2009*). We found no obvious differences in a

crystal structure of this mutant in the closed conformation (*Figure 5—figure supplement 1c,d* and *Supplementary file 2*). Overall, a significant contribution to K$^+$ selectivity by the extracellular ion binding site is contradicted by the results for the T38A mutant protein - whether the residual selectivity is arising from this motif remains open. Ascribing a function to the extracellular ion binding site is thus currently difficult. Structural insight into the conductive conformation, which might reveal rearrangements at both ends of helix 1, will help to gain insight into potential functions of this region. The binding site could also serve an unrelated function, for example sensing of ions to modulate the open-probability.

## A layer of threonines also accounts for K$^+$ selectivity in human TMEM175 channels

The layer of threonines is also conserved in vertebrate TMEM175 proteins (*Figure 5c*) and we thus tested if mutating the corresponding residues to alanine in the human TMEM175 channel would affect selectivity. Human TMEM175 is more selective than the bacterial counterparts with reported values of P$_K$/P$_{Na}$ of 35-20 (*Cang et al., 2015*; *Lee et al., 2017*). We measured currents from cells expressing hTMEM175 at the plasma membrane as previously done (*Lee et al., 2017*). From the shift in the reversal voltage after replacing K$^+$ for Na$^+$ in the external medium ($-58 \pm 3$, n = 7) we estimate a P$_K$/P$_{Na}$ value of ~10 (*Figure 5d*), somewhat lower than the previously reported P$_K$/P$_{Na}$ values of hTMEM175. Mutating Thr49 in the first repeat and Thr274 in the second repeat of hTMEM175 to alanine resulted in strongly reduced selectivity with a reversal potential of $-17 \pm 3$ (n = 8) when exchanging K$^+$ for Na$^+$ in the external solution (*Figure 5d*), providing evidence for a conserved role of the threonine-layer in selectivity. Since the human channel is 2–3 times more selective as the known bacterial counterparts there must thus be an additional factor that accounts for the increased selectivity, probably in conjunction with the threonine layer.

## Serine45 increases K$^+$ selectivity in human TMEM175

When comparing the primary sequences of vertebrate genes, we found that the position that is corresponding to the highly conserved Ala34 in MtTMEM175 is occupied by serine in repeat one for all of the analyzed vertebrate species (Ser45 in hTMEM175), while in repeat 2, like in bacterial channels, the corresponding residue is an alanine (Ala270 in hTMEM175) (*Figure 5c* and *Figure 5—figure supplement 1e,f*). We have thus suspected that Ser45 might also play a role for selectivity in hTMEM175 in an analogous manner as the threonines. The side chain of these serines could contribute to coordination of K$^+$ ions in close proximity to the threonine layer to increase selectivity, that is six ligands would be involved in ion coordination instead of only four as in MtTMEM175. In the S45A mutant dimer, selectivity was indeed reduced with a reversal potential of $-27 \pm 2$ (n = 24) upon changing the major cation in the bath solution from K$^+$ to Na$^+$. This mutant is very similar to bacterial TMEM175 channels in its primary sequence of helix one and intriguingly also with respect to its preference for K$^+$. Consequently, a triple mutant where all of the threonine and serine residues in these two layers of the pore are exchanged for alanine (S45A/T49A/T274A) shows a similar reduction of selectivity (E$_{rev}$ = $-22 \pm 3$ mV, n = 8) (*Figure 5d*) as the double mutant T49A/T249A and the T38A mutant protein of MtTMEM175 (*Figure 5b*). We thus conclude that Ser45 in the first repeat is accounting for the increased selectivity of the human TMEM175 channel, but in conjunction with the threonine layer.

As the total number and geometry of the coordinating ligands accounts for selectivity, it is not surprising that mutation of the threonine layer suffices to lose selectivity since the remaining two serine residues alone could not effectively coordinate K$^+$ ions. Generally, reduction of sequential ion binding sites is known to attenuate the K$^+$ selectivity in the canonical K$^+$ selectivity filter, whereas introducing additional binding sites can increase selectivity (*Derebe et al., 2011*; *Sauer et al., 2013*; *Kast et al., 2011*; *Lee and MacKinnon, 2017*; *Alam and Jiang, 2009b*; *Gouaux and Mackinnon, 2005*), a principle that apparently also accounts for the different K$^+$ selectivity of bacterial and vertebrate TMEM175 channels.

## Zinc ions and 4-aminopyridine act as pore blockers at the selectivity filter of hTMEM175

Next, we sought to gain insight into the mechanism of channel blocking in TMEM175 proteins. In comparison to bacterial TMEM175 channels, the human TMEM175 channel is substantially more sensitive to $Zn^{2+}$ ions (IC50 ~38 µM compared to an estimated IC50 of ~0.5 mM for bacterial homologues [*Cang et al., 2015*]). In addition, the human channel is also inhibited by the potassium-channel blocker 4-AP (IC50 ~35 µM) (*Cang et al., 2015*), indicating significant differences between human and bacterial homologues and suggesting a more specific interaction of the blockers with the human channel. The equally effective block of human TMEM175 by $Zn^{2+}$ and 4-AP regardless of extracellular or intracellular application (*Cang et al., 2015*; *Lee et al., 2017*) is a good indication that the block is occurring in the pore. We have shown that the increased selectivity for $K^+$ in hTMEM175 is founded on Ser45, and thus assumed that the pronounced sensitivity for $Zn^{2+}$ and the potency of 4-AP could also be based on this difference. To address this question, we analyzed the response of the S45A mutant of hTMEM175 for these blockers in comparison with the WT protein. As shown in *Figure 6a and b*, the S45A mutant is not blocked by $Zn^{2+}$ ions and also lost its sensitivity for 4-AP at a concentration of 100 µM.

These data confirm that both $Zn^{2+}$ and 4-AP act as pore blockers at the selectivity filter, at very similar locations. The size of the 4-AP molecule further suggests that this block can only take place in a widened pore which again implies a different conductive conformation where bulky side chains do not occlude the ion path and Ser45 is facing the pore.

Bacterial channels are only weakly inhibited by $Zn^{2+}$(*Cang et al., 2015*) and the S45A mutant of hTMEM175 suggests that the threonine layer constituted of Thr38 in MtTMEM175 and Thr49/Thr274 in hTMEM175 does not suffice to render TMEM175 channels sensitive to $Zn^{2+}$. Indeed, the T38A mutant of MtTMEM175 retained its sensitivity for $Zn^{2+}$; in WT and the mutant channel $Zn^{2+}$ caused at a reference voltage of −60 mV the same relative inhibition (WT 66.9 ± 12%, T38A mutant 69.7 ± 10 n = 4) (*Figure 6—figure supplement 1a*). This suggests that the bacterial channel is

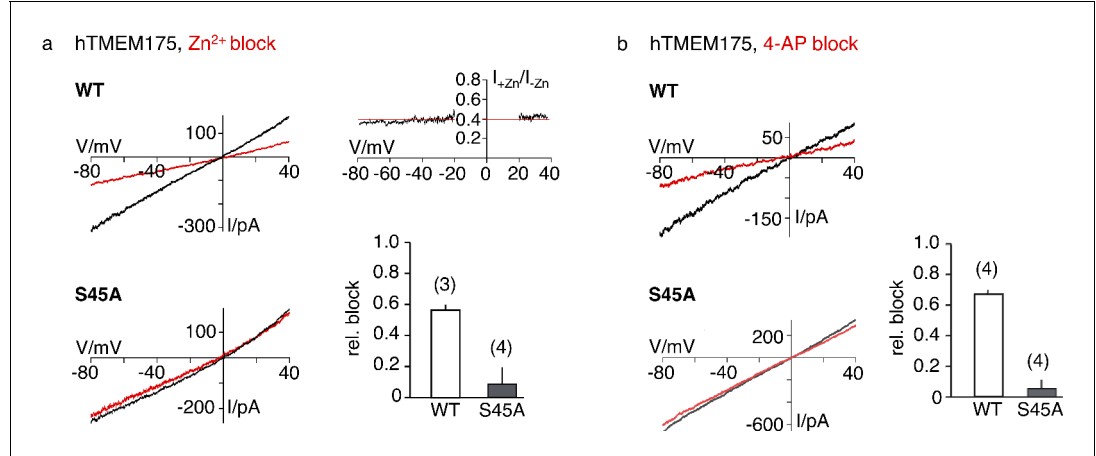

**Figure 6.** Sensitivity of the hTMEM175 S45A mutant for $Zn^{2+}$ and 4-AP. (a) Currents elicited by a ramp protocol (−80 to +40 mV in 200 ms) in HEK293 cells expressing hTMEM175 WT (upper left) or hTMEM175 S45A mutant (lower left) in absence (black) and presence (red) of 5 mM $ZnSO_4$ in external bath solution (150 mM $K^+$). Columns (lower right) summarize average inhibition (± s.d.) of current amplitudes at −60 mV from 3 and 4 recordings in the hTMEM175 WT and S45A mutant, respectively. The ratio of currents in the presence and absence of $Zn^{2+}$ (I+Zn/I-Zn) show the voltage independence of channel block (upper right). (b), same as in (a) with representative measurements in absence (black) or presence (red) of 100 µM 4-AP in external bath solution (150 mM $K^+$) for hTMEM175 WT (upper left) and S45A mutant (lower left), respectively. Columns show average inhibition (± s.d.) of current amplitudes at −60 mV from four measurements in the hTMEM175 S45A mutant and WT, respectively.

The online version of this article includes the following source data and figure supplement(s) for figure 6:

**Source data 1.** Raw data for *Figure 6a*.
**Source data 2.** Raw data for *Figure 6b*.
**Figure supplement 1.** Sensitivity to $Zn^{2+}$ and anomalous density of Zn in the pore of MtTMEM175.
**Figure supplement 1—source data 1.** Raw data for *Figure 6—figure supplement 1*.
**Figure supplement 2.** Constriction points in MtTMEM175 and CmTMEM175.

inhibited by $Zn^{2+}$ through a different mechanism, that is not a pore block. In seeming contrast to this finding we could clearly localize anomalous signal of $Zn^{2+}$ ions in the pore by soaking crystals of MtTMEM175 with 0.5 mM $Zn^{2+}$. The major peak of this anomalous signal (when contoured at 4σ) lies in proximity to Thr38 and Leu35 (*Figure 6—figure supplement 1b*). In crystal soaking experiments with the T38A mutant of MtTMEM175 we could not detect anomalous signal for Zn, even when using concentrations of 2.5 mM (0.5–2.5 mM tested) and contouring of the anomalous signal at 4σ and 2σ (*Figure 6—figure supplement 1c*). This supports an attraction for $Zn^{2+}$ ions in the pore in close distance to, or directly at the selectivity filter and indicates that the T38A mutation changes the electrostatic environment in the pore. However, our functional analysis does not support the idea that $Zn^{2+}$ is blocking MtTMEM175 by the same mechanism as hTMEM175.

## Discussion

The TMEM175 family of non-canonical potassium channels has recently been identified to confer a $K^+$ selective conductance to lysosomes and late endosomes (*Cang et al., 2015*). Importantly, it has been shown that this channel is presumably involved in the early onset of PD (*Jinn et al., 2017*; *Nalls et al., 2014*; *Chang et al., 2017*; *Jinn et al., 2019*). Even though its exact function in lysosomal physiology still needs to be clarified the available experimental evidence shows that aberrant processing of autophagosomes as well as an increased lysosomal pH under conditions of starvation is the prominent phenotype of TMEM175-loss (*Cang et al., 2015*; *Jinn et al., 2017*) and likely connects this lysosomal pathology to PD. In combination with an electrophysiological analysis the high-resolution MtTMEM175 structure provides a solid framework on which we identify the residues that confer $K^+$ selectivity in this channel family.

### Selectivity in TMEM175 channels

From the structural analysis we divide the pore of TMEM175 channels into functional layers (*Figure 7a*), conceptually different from a previous interpretation (*Lee et al., 2017*): The TMEM175 ion pathway is built from an ion binding site for monovalent cations (with properties similar to $K^+$ ions) at the extracellular pore entrance, a major gate at the position of Leu35 (MtTMEM175) and, rather unusual, by one (in prokaryotes) or two interspersed polar layers (in vertebrates) that tune $K^+$ selectivity. Our observations and conclusions are based on two important considerations. First, scrutiny of the pore in the structure of MtTMEM175 shows that it is too narrow for the passage of ions. This demands that conformational changes have to take place in order to make the channel conductive. Second, there are highly conserved hydrophilic side chains (from threonine and additionally serine in vertebrate counterparts) that would be suited for coordinating ions on their passage. But since these residues do not face the pore lumen we anticipate a rotation of helix 1, resulting in an iris-like opening in assembled TMEM175 channels as a plausible route for transitioning into a conductive conformation. Such a rotation simultaneously exposes the hydroxyl-groups of the threonines (Thr38 in MtTMEM175 and Thr49/Thr274 in hTMEM175) and serines (Ser45 in hTMEM175) to the pore lumen and swings the bulky hydrophobic residues (Leu35 in MtTMEM175) out of the conducting pathway (*Figure 7b*). Indeed, mutating the respective threonine and serine residues strongly reduced the $K^+$ selectivity both in bacterial and vertebrate TMEM175 channels (*Figure 5b,d*).

Intriguingly, in hTMEM175 the selectivity could be attenuated to levels of bacterial homologues by reducing the number of coordinating ligands from six to four through mutation of Ser45 to alanine. How exactly the presence of these two additional hydrophilic residues leads to a preference for $K^+$ over $Na^+$ ions remains to be seen. But as previously mentioned, a higher number of coordinating ligands seems to be a general mechanism for achieving a high $K^+$ selectivity also in other $K^+$ channels (*Derebe et al., 2011*; *Sauer et al., 2013*; *Kast et al., 2011*; *Lee and MacKinnon, 2017*; *Alam and Jiang, 2009b*; *Gouaux and Mackinnon, 2005*).

The proposed mechanism for $K^+$ selectivity in TMEM175 channels is also backed by our finding that Ser45 is underlying a pore block by $Zn^{2+}$ and 4-AP in hTMEM175 at the site of the selectivity filter, underscoring that the respective Ser residues are part of the conductive pathway. The data further suggests why bacterial channels are less sensitive or resistant to these agents with only four ligands in their simpler selectivity filter.

Another unusual finding is the voltage independence of the pore block in hTMEM175 (*Figure 6a*), as this is expected for a binding site in the electric field of the channel (*Hilf et al.,*

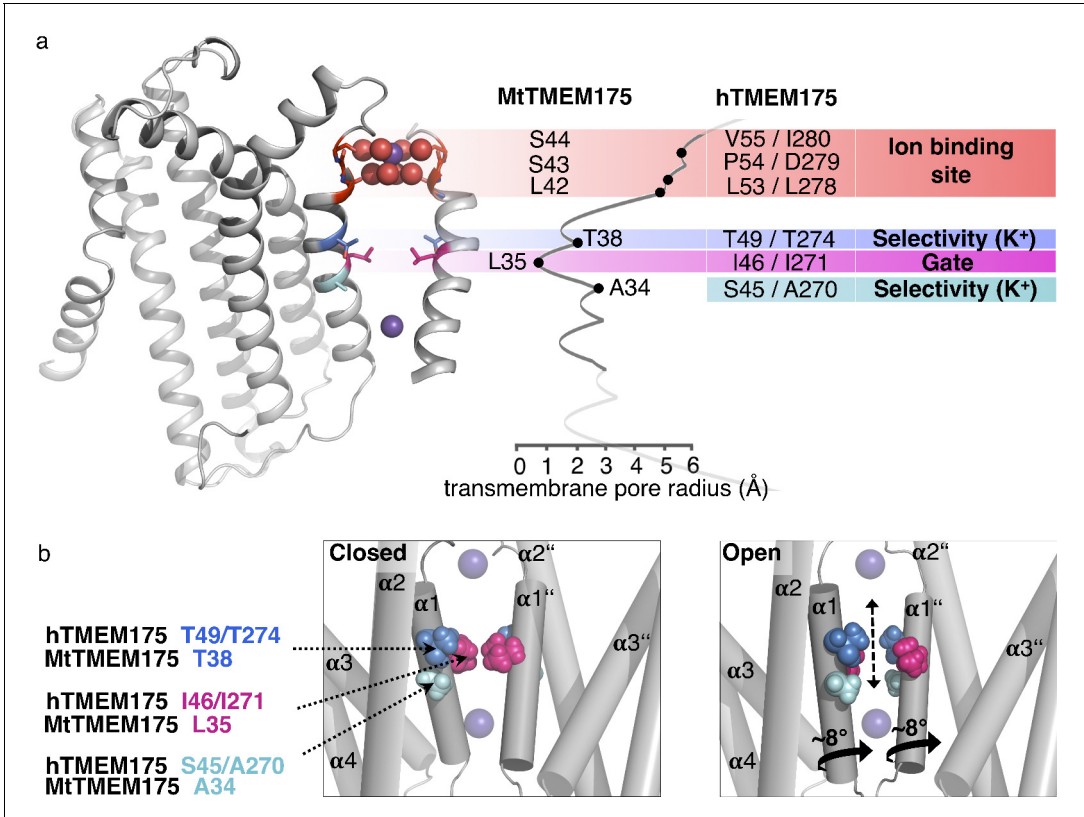

**Figure 7.** Functional layers and proposed mechanism for channel opening in TMEM175 channels. (**a**) Functional layers in the MtTMEM175 pore. Two subunits are shown (right side only partly). Important residues in MtTMEM175 and hTMEM175 and for comparison the pore radius (in Å) are indicated. The ion binding site is shown in red, gate-residues in magenta and residues required for selectivity in blue and cyan. (**b**) Schematic side view of MtTMEM175 in closed (left) and proposed conductive state (right). Key residues on helix one are shown as spheres. Two subunits are omitted for clarity. $K^+$ ions are shown as purple spheres. A clockwise rotation of helix 1 (viewed from intracellular) of 8–15° would widen the pore sufficiently for $K^+$ ions to permeate (indicated by curved arrows in the bottom panel).

*2010*). How can this be explained? We can currently only speculate but one reason could be that TMEM175 channels spontaneously open and close without a trigger, different from voltage- or ligand-gated ion channels. When they are open, the binding site for the pore blocker is accessible and the blocker could bind. It is worth to note that in our crystal soaks the small $Zn^{2+}$ ion was found even in the closed channel nearby the selectivity filter in the middle of the membrane plane (*Figure 6—figure supplement 1b*). This could mean that the $Zn^{2+}$ ion (or the 4-AP molecule) is already bound at the selectivity filter before a voltage is applied for eliciting a current. In such a scenario voltage may not affect the strength of the block.

In summary, our data advocate a model which provides a coherent description of selectivity, blocking and channel opening. However, definite conclusions have to await additional evidence from an open channel structure.

## Structural differences between MtTMEM175 and CmTMEM175

Comparison of the previously reported structure of CmTMEM175 (*Lee et al., 2017*), with the model of MtTMEM175 highlights a number of significant differences, which potentially bear information on the function of TMEM175 type channels.

One crucial difference is located at the cytosolic entrance to the pore and could be related to a pH dependent gating mechanism. While both structures reveal a hydrogen-bond network on the intracellular side within helices 1–3, only the MtTMEM175 structure exhibits an additional interaction between His77 with Arg24, which connects two adjacent subunits. The corresponding arginine in CmTMEM175 is in contrast not clearly resolved (*Figure 1—figure supplement 1g*). Since

MtTMEM175 was crystallized at a pH of 8.5 and CmTMEM175 at a pH of 4.5 it is most likely that His73 in CmTMEM175 (corresponding to His77 in MtTMEM175) is protonated and hence repulsing the arginine. A transient pH-sensitive disengagement of this arginine from the histidine (and Asp30) could be a step in the opening of the channel. This would be achieved by loosening interactions of the N-terminal end of helix one with the adjacent subunit as already indicated in the CmTMEM175 structure which shows a more open pore at the intracellular side compared to MtTMEM175 (*Figure 6—figure supplement 2c*).

Another difference between the structures is apparent at the extracellular tip of helix one where both structures are substantially deviating. This can explain the lack of coordinated ions in the CmTMEM175 structure (*Figure 3—figure supplement 1i,k*). Notably, only the extracellular end of helix one in the CmTMEM175 structure contains a $3_{10}$-helix over a stretch of 3–4 residues and thus extends further than helix one in MtTMEM175 (*Figure 3—figure supplement 1f,g*). As a consequence the backbone oxygens are too distant from the pore axis for interacting, like in the MtTMEM175 structure, with the water molecules of a $K^+$-hydrate (*Figure 3a,b,d* and *Figure 3—figure supplement 1i,k*). $3_{10}$-helices are commonly associated with transition states and thus hint to dynamic regions within a helix (*Vieira-Pires and Morais-Cabral, 2010*). Whether the CmTMEM175 structure represents such an intermediate conformation (albeit likely not a true transition state) remains to be investigated.

It is furthermore worth noting, that the short helix between the helices 1 and 2 of CmTMEM175 is involved in major crystal contacts, which might be responsible for a displacement of the tips of helix1 and formation of the $3_{10}$-helix (*Figure 3—figure supplement 1h*). Even though our results indicate that the coordinated $K^+$ ion has no direct impact on the selectivity of the MtTMEM175 channel, this configuration may still be relevant for function. However, the $3_{10}$-helix in CmTMEM175 has also an influence on the position of other pore lining side chains when compared to MtTMEM175. This is in particular true for Leu30, which is facing the pore in contrast to the equivalent Leu42 in MtTMEM175 (*Figure 6—figure supplement 2a,b*). Leucine30, Leu27 and Ile23 in CmTMEM175 are thereby forming three layers of pore-lining hydrophobic residues. This scenario is reminiscent of the structures of bestrophins, for example a channel to which CmTMEM175 was primarily compared in literature (*Lee et al., 2017*). It was proposed that Ile23, which is part of the triad of bulky pore lining residues in CmTMEM175, conveys the $K^+$ selectivity as a hydrophobic selectivity filter, as initially also proposed for bestrophins (*Yang et al., 2014*; *Kane Dickson et al., 2014*). Mutation of Ile23 or corresponding residues in hTMEM175 to small or hydrophilic residues indeed resulted in a loss of $K^+$ selectivity (*Lee et al., 2017*). If Ile23 however acts as a gate to keep the channel closed, a role that we suggest for the corresponding Leu35 in MtTMEM175, an exchange for asparagine or alanine would likely result in a permanently open channel (*Figure 6—figure supplement 2d*). For instance, in bestrophins, mutation of three layers of bulky residues along the ion path to alanine resulted in an open channel, without the requirement for activation (*Vaisey et al., 2016*; *Rao et al., 2017*). This supported a function of these residues as a gate instead of contributing to selectivity. Similarly, mutation of a gate built from phenylalanine in the NaK channel strongly increased flux (*Alam and Jiang, 2009a*). More recently a cryo-EM structure of chicken bestrophin in a conductive conformation provided evidence against a role of the bulky pore-occluding residues as hydrophobic selectivity filters but instead revealed that these residues are indeed physical gates (*Miller et al., 2019*). Hence mutation of hydrophobic gates with small and/or hydrophilic residues can have substantial impact on the closure and conduction of ion channels (*Rao et al., 2017*; *Rao et al., 2018*; *Aryal et al., 2014*). When we mutated Leu35 to alanine we could only see small effects on the selectivity, speaking against a role in selectivity (*Figure 5b*). Probably this mutant allows for some non-specific permeation in closed state due to the loss of the leucine-gate which results in slightly reduced selectivity in whole-cell recordings when compared to WT.

## Conclusions

TMEM175 channels are not as enigmatic as anticipated earlier but instead recapitulate classical structure-function correlates of other ion channel families: Large hydrophobic residues are acting as gates and polar contacts from side chains and the backbone are coordinating ions in the conducting pathway. On the other hand, it is remarkable that the selectivity is mediated by cryptic hydroxyl-bearing side chains inside the pore that are only available for selective ion solvation in an open conformation by concerted repositioning. In conclusion, the conductive state must thus deviate

substantially from the closed state in order to be permeable to ions. This is strongly supported by the localization of the residues that mediate K⁺ selectivity and sensitivity to the blockers $Zn^{2+}$ and 4-AP. Collectively this study provides insight into an alternative solution for conduction of K⁺ ions and an unusual selectivity filter. But with respect to the geometry, chemistry and the number of ligands the channel architecture also recapitulates established concepts of K⁺ channel biophysics.

# Materials and methods

## Key resources table

| Reagent type (species) or resource | Designation | Source or reference | Identifiers | Additional information |
|---|---|---|---|---|
| Gene (*Marivirga tractuosa*) | MtTMEM175 | DSM 4126 | E4TN31 | |
| Gene (*Homo sapiens*) | hTMEM175 | Sourcebioscience | Q9BSA9, IRAUp969F1019D | |
| Gene (*Streptomyces collinus*) | scTMEM175 | Synthesized by GenScript | S5VBU1 | |
| Recombinant DNA reagent | pBXC3H | Dutzler lab | Addgene # 47068 | |
| Recombinant DNA reagent | pcDXC3M | Dutzler lab | Addgene #49030 | |
| Recombinant DNA reagent | pcDXC3GMS | Dutzler lab | Addgene #49031 | |
| Recombinant DNA reagent | pBXNPHM3 | Dutzler lab | Addgene #110099 | |
| Cell line (*Homo sapiens*) | HEK293 | Germ. Collection Microorg. & cell cultures, Braunschweig, Germany | ACC 305, tested mycoplasma negative | |
| Strain, strain background (*Escherichia coli*) | MC1061 | Dutzler lab | Made by Malcolm Casadaban https://cgsc.biology.yale.edu/Strain.php?ID=11225 | |
| Commercial assay or kit | Superdex 200 10/300 | GE healthcare | Cat#17517501 | |
| Commercial assay or kit | Pierce Streptavidin Plus UltraLink resin | Thermo Fisher Scientific | Cat#53117 | |
| Commercial assay or kit | Ni-NTA resin | Qiagen | Cat#30230 | |
| Commercial assay or kit | Superdex 200 5/150 increase | GE healthcare | Cat#28990945 | |
| Commercial assay or kit | Strep-Tactin Superflow high capacity 50% suspension | Iba lifesciences | Cat#2-1208-010 | |
| Chemical compound, drug | cOmplete, EDTA-free Protease Inhibitor Cocktail | Roche | Cat#5056489001 | |

*Continued on next page*

*Continued*

| Reagent type (species) or resource | Designation | Source or reference | Identifiers | Additional information |
|---|---|---|---|---|
| Chemical compound, drug | n-dodecyl-β-d-maltopyranoside, Solgrade | Anatrace | Cat#D310S | |
| Chemical compound, drug | n-decyl-β-d-maltopyranoside, Solgrade | Anatrace | Cat#D322S | |
| Chemical compound, drug | Desthiobiotin | Iba lifesciences | Cat#2-1000-002 | |
| Chemical compound, drug | *E. coli* polar lipids | Avanti polar lipids | Cat#100600 | |
| Chemical compound, drug | Fugene | Promega | Cat#E2311 | |
| Chemical compound, drug | CellMask Deep Red | Thermo Fisher | Cat#C10046 | |
| Chemical compound, drug | ER-tracker Blue-white DPX | Thermo Fisher | Cat#E12353 | |
| Chemical compound, drug | Accutase | Thermo Fischer | Cat#A1110501 | |
| Chemical compound, drug | Dulbecco's modified Eagle's medium | Sigma | Cat#D5671 | |
| Chemical compound, drug | L-Glutamine | Sigma | Cat#G7513 | |
| Chemical compound, drug | GeneJuice Transfection Reagent | Millipore Corp | Cat#70967-5 | |
| Chemical compound, drug | Fetal bovine serum (FBS) | Sigma | Cat#BCBV7601 and Cat#F7524 | |
| Other | EPC-9 Amplifier | Heka Electronics | | |
| Software, algorithm | Fiji | *Schindelin et al., 2012* | http://imagej.net/Fiji | |
| Software, algorithm | PulseFit | Heka Electronics | | |
| Software, algorithm | PatchMaster V2x90,3 | Heka Electronics | | |
| Software, algorithm | FitMaster V2x90,1 | Heka Electronics | | |
| Software, algorithm | IGOR version 6.3.7.2 | WaveMetrics | | |
| Software, algorithm | JPCalcWin version 1.01 | https://medicalsciences.med.unsw.edu.au/research/research-services/ies/jpcalcwin | | |

## Cloning

Thirty TMEM175 genes were cloned from genomic DNA of various eubacteria. The genes were flanked by a 3C protease cleavage site, a myc-tag and a StrepTagII, either on the N- or C-terminus. The TMEM175 gene of *Marivirga tractuosa* (UniProt accession # E4TN31) was cloned from the strain DSM 4126. The TMEM175 cDNA of *Streptomyces collinus* (UniProt accession # S5VBU1) was synthesized by GenScript. For expression in MC1061 *E. coli*, TMEM175 genes were expressed from the FX-cloning plasmid pBXC3H (*Geertsma and Dutzler, 2011*) (Addgene # 47068) with a stop codon. For electrophysiology and expression in HEK293 cells, the TMEM175 genes were cloned without tags into the plasmids pcDXC3MS (*Brunner et al., 2014*; *Schenck et al., 2017*) (Addgene #49030) followed by a stop-codon as well as into the vector pcDXC3GMS (*Brunner et al., 2014*; *Schenck et al., 2017*) (Addgene #49031) (where EGFP was replaced by Venus-YFP (vYFP) using the *KpnI* sites) to obtain a C-terminally vYFP tagged channel. Both, tagged and untagged versions yielded similar results. For TIRF microscopy the TMEM175 genes were cloned into the vector pcDXC3GMS

(*Brunner et al., 2014*; *Schenck et al., 2017*) (Addgene #49031) to produce a fusion to vYFP. For cell surface labeling of MtTMEM175 with Nb$_{51H01}$ and for size exclusion profiles of MtTMEM175 expressed in HEK cells, MtTMEM175 was cloned into pcDXC3MS (*Brunner et al., 2014*; *Schenck et al., 2017*) (Addgene #49030) to include a streptavidin binding peptide tag for purification (SBP) (*Keefe et al., 2001*). Nb$_{51H01}$ was cloned into pcDXC3GMS (Addgene #49031) to include a vYFP tag in addition to the purification tag. For the selection of nanobodies, the MtTMEM175 gene was cloned into pBXC3H to purify the protein using a deca- histidine tag. An Avi-Tag for biotinylation was introduced by PCR preceding the histidine tag at the C-terminus. Positive nanobodies were subcloned into the plasmid pBXNPHM3 (*Schenck et al., 2017*; *Ehrnstorfer et al., 2014*; *Geertsma et al., 2015*) (Addgene #110099) for expression. C-terminally MBP (malE, *Escherichia coli K12*) tagged versions of nanobodies were generated by cloning nanobody genes and N-terminally truncated MBP genes into pBXNPHM3. The last four amino acids of MBP (RITK) were replaced with PG. The resulting expression construct consists of a nanobody, a valine linker that connects the N-terminally truncated MBP, preceded by the pelB leader sequence, a deca- histidine tag, an MBP and a 3C protease cleavage site as depicted in *Figure 1—figure supplement 1b*. Mutant proteins were generated by site directed mutagenesis. All constructs were verified by Sanger sequencing.

## Cell culture and transfection protocol

Membrane currents were recorded from HEK cells transiently expressing TMEM175 proteins. For this low passage human embryonic kidney (HEK293) cells were cultured in Dulbecco's modified Eagle's medium supplemented with 10% fetal bovine serum, 100 IU/mL of penicillin, 100 μg/ml of streptomycin, and stored in a 37°C humidified incubator with 5% $CO_2$. Transfections were performed with GeneJuice Transfection Reagent (Millipore Corp) according the producer protocol: The MtTMEM175 genes inserted in pcDXC3MS (*Brunner et al., 2014*; *Schenck et al., 2017*) were co-transfected with a plasmid containing green fluorescent protein (GFP) and incubated in dark. Human TMEM175 constructs were cloned into pcDXC3GMS (*Brunner et al., 2014*; *Schenck et al., 2017*) with a C-terminal vYFP tag.

## Patch clamp recordings

One to two days after transfection, cells were dispersed by accutase treatment and seeded on 35 mm plastic petri dishes (on 15 mm cover slips) to allow single cell measurements. Green fluorescent cells were selected for patch clamp measurements. Membrane currents were recorded in whole cell configuration using an EPC9 or EPC10 patch-clamp amplifier (HEKA Electronics) controlled by the PatchMaster software (HEKA). Micropipettes with a resistance of about 2 MΩ were made from 1.5 mm thin-walled glass and fire-polished. The pipette solution contained (in mM) 150 KOH, 5 HCl, 10 HEPES, pH 7.4, titrated with methanesulfonic acid. The standard bath solution contained (in mM) 150 KOH, 1 $CaCl_2$, 1 $MgCl_2$, 10 TEA, 10 HEPES/KOH, pH 7.4, titrated with methanesulfonic acid. For measurements of selectivity $K^+$ was replaced by other cations of interest. Relevant liquid junction voltages were calculated with JPCalcWin (UNSW Sydney). Differences in osmolarity between pipette and bath solution were compensated by D-mannitol. Membrane currents were either measured by voltage step- or ramp protocols. In standard step protocol the cell was clamped for 200 ms in 20 mV steps from holding voltage (0 mV, 100 ms) to test voltages between ±100 mV before returning to holding voltage (100 ms). The steady state current at the test voltages was measured during the final 20 ms of clamp steps. In ramp protocols the voltage increased from a holding voltage (−80 mV, 20 ms) in 200 ms to +40 mV (20 ms).

## Expression and purification of MtTMEM175 from *E. coli*

MC1061 *E. coli* cells harboring the C-terminally tagged MtTMEM175 gene were grown in terrific broth with 100 μg/ml ampicillin to an OD$_{600}$ of 0.5 at 37°C. Expression was induced with 0.02% Arabinose and continued over night at 18°C. Cells were harvested and resuspended in 150 mM NaCl, 50 mM Hepes-NaOH pH 7.6, 10% glycerol containing protease inhibitors (Complete, Roche), DNase I and 5 mM $MgCl_2$. Cells were lysed at 15000–25000 p.s.i. Cell debris was removed by centrifugation at 8000 g for 30 min. Membranes in the supernatant were harvested by centrifugation using a 45 TI rotor (Beckmann) at 42000 r.p.m. for 1 hr and resuspended in 250 mM KCl, 20 mM Hepes-NaOH pH 7.6, 15% glycerol. Extraction of the protein was carried out using 2% *n*-dodecyl-β-d-maltopyranoside

(DDM, Anatrace) and protease inhibitors (Roche) for 1 hr and subsequently centrifuged at 42000 r. p.m. using a 45 Ti rotor (Beckmann). The supernatant was incubated in batch with Strep-Tactin resin (Strep-Tactin Superflow high capacity, iba/Göttingen) for 1 hr, washed with 150 mM KCl, 10 mM Hepes-NaOH pH 7.6, 10% glycerol, 50 µg ml$^{-1}$ *E. coli* polar lipids (Avanti) and 0.03% DDM, and MtTMEM175 was eluted with the wash buffer containing 5 mM *d*-Desthiobiotin (Sigma-Aldrich). The protein was cleaved using HRV 3C protease and concentrated to 10–20 mg/ml using Amicon concentrators (Millipore) with a 100 kDa cutoff. The MtTMEM175 protein was mixed with $Mb_{51H01}$ in a molar ratio of 2.2–2.5. For this, concentrated $Mb_{51H01}$ was supplemented with 3 mM maltose to keep MBP in the substrate-bound conformation and DDM was added to 0.03%. After that, concentrated MtTMEM175 was added for complex formation. The mixture was left on ice for 30 min and applied to a Superdex 200 10/300 column (GE healthcare) equilibrated with 150 mM KCl, 5 mM Hepes-NaOH pH 7.6, 2.5 mM Maltose and 0.03% DDM. The peak fractions were pooled and concentrated to 8–16 mg/ml for crystallization. All Steps were performed on ice or at 4°C. Mutant proteins were purified in the same way.

## Expression and purification of MtTMEM175/fluorescent $Nb_{51H01}$ from HEK293 cells

Low passage HEK293 cells were transiently transfected with MtTMEM175 (cloned into pcDXC3MS [*Brunner et al., 2014*; *Schenck et al., 2017*], Addgene #49030) or $Nb_{51H01}$ (cloned into pcDXC3GMS [*Brunner et al., 2014*; *Schenck et al., 2017*], Addgene #49031) using Fugene following the manufacturers protocol. 40 hr after transfection, the cells were harvested. The proteins containing a streptavidin-binding peptide (SBP) tag (*Keefe et al., 2001*) were purified using Pierce Streptavidin Plus UltraLink resin as described (*Brunner et al., 2014*; *Schenck et al., 2017*) but using the same buffers as described for the purification of MtTMEM175 from *E. coli* expressions, except that 5 mM *d*-desthiobiotin in the elution buffer was replaced with 3 mM biotin. Size exclusion chromatography was performed using a Superdex200 increase 5/150 column.

## Multi angle laser light scattering (MALLS) measurements

3C-protease cleaved MtTMEM175 protein was purified as described above except that the peak fraction after size exclusion chromatography was diluted to 35 µM (1 mg/ml) before subjecting it to MALLS-SEC using a Superdex 200 10/300 column (GE healthcare) with an Agilent LC-1100 system coupled to an Optilab rEX refractometer (Wyatt Technology) and a miniDAWN 3-angle light-scattering detector (Wyatt Technology). The SEC buffer contained 150 mM KCl, 10 mM Hepes-NaOH and 0.03% DDM at pH 7.6 at RT. Data was analyzed with ASTRA software (Wyatt Technology).

## Generation of nanobodies in alpacas

Nanobodies against MtTMEM175 were raised in alpacas (*Vicugna pacos*) at the Nanobody Service Facility of the University of Zurich, NSF/UZH) as previously described (*Schenck et al., 2017*). Briefly, alpacas were immunized four times with 14 day intervals by injecting 100 µg of purified MtTMEM175 protein at a concentration of 35 µM (in 150 mM KCl, 10 mM Hepes-NaOH pH 7.6, 0.03% DDM, 15% glycerol) subcutaneously. A blood sample was used to generate lymphocyte cDNA by reverse transcription. Nanobody genes were cloned into a phagemid vector to create a phage library which was screened by biopanning against biotinylated MtTMEM175 immobilized on Neutravidin-coated plates. Biotinylation was performed as described using recombinant BirA enzyme (*Ehrnstorfer et al., 2014*; *Geertsma et al., 2015*). Positive binders were identified using ELISA and subcloned into pBXNPHM3 for expression.

## Expression and purification of nanobodies/macrobodies

For expression of nanobodies in the vector pBXNPHM3, MC1061 *E. coli* cells were grown to an $OD_{600}$ of 0.75 at 37°C in terrific broth containing 100 µg/ml ampicillin. Protein expression was started by addition of 0.02% Arabinose and continued for 3.5 hr at 37°C. Cells were harvested and resuspended in 150 mM NaCl, 50 mM Tris-HCl pH 8, 20 mM imidazole, 5 mM $MgCl_2$, 10% glycerol, 10 µg/ml DNAse I and protease inhibitors (Complete, Roche). Cells were lysed at 15000–25000 p.s.i. Cell debris was removed by centrifugation at 42000 r.p.m in a 45 Ti rotor. The supernatant was applied in batch to NiNTA-resin for 1 hr, washed with 150 mM KCl, 40 mM imidazole pH7.6, 10%

glycerol and eluted with 150 mM KCl, 300 mM imidazole pH 7.6, 10% glycerol. The protein was cleaved over-night using HRV 3C protease during dialysis against 150 mM KCl, 10 mM Hepes-NaOH, 20 mM imidazole, pH 7.6, 10% glycerol. The MBP–His$_{10}$-fragment was removed by binding to NiNTA resin and the flow-through containing the nanobodies was concentrated (Amicon) and applied to a Superdex 200 column (GE healthcare) equilibrated in 150 mM KCl, 5 mM Hepes 7.6. The peak fractions were concentrated to 10–25 mg/ml before mixing with MtTMEM175 for complex formation. Complex formation of purified nanobodies with MtTMEM175 was analyzed by SEC, where Nb$_{51H01}$ (corresponding macrobody is Mb$_{51H01}$) was identified as a MtTMEM175 binder with a 1:1 stoichiometry. Macrobodies were expressed and purified in the same way.

## Crystallization of the MtTMEM175-Mb$_{51H01}$ complex

Expression and monodispersity of purified TMEM175 proteins in small scale was analyzed by SDS-PAGE and SEC. Several TMEM175 proteins were expressed at reasonable rates and eluted as mono-disperse species from SEC. Expression was scaled up and we could crystallize several homologues readily. However, all of the crystallized proteins, including MtTMEM175, diffracted not beyond 20 Å, even after extensive optimization of the crystallization conditions. To improve crystallization, we generated nanobodies against MtTMEM175 as described above. Nb$_{51H01}$, identified by ELISA and SEC, was used for complex formation with MtTMEM175 and this complex was subjected to crystallization. The best crystals of this complex diffracted not beyond 10 Å. To improve crystallization further we decided to fuse MBP to the C-terminus of Nb$_{51H01}$ in order to increase possible crystal contacts and the chance for advantageous crystal lattices. We fused the Nb at the C-terminus with an N-terminally truncated MBP (starting at Lys [*Cang et al., 2015*] without the signal sequence) linked by a valine residue as depicted in *Figure 1—figure supplement 1b*. This resulted in the interfacial sequence PVTV**V***KLVIWIN* (Nb C-terminus underlined, linker in bold and MBP N-terminus in italics) and we named the construct Mb$_{51H01}$. A complex of Mb$_{51H01}$ and MtTMEM175 was purified by SEC. Before subjecting the sample to SEC the mixture was left on ice for 15–30 min and eluted in 150 mM KCl, 5 mM Hepes-NaOH, 2.5 mM maltose and 0.03% DDM. The fractions containing the complex were concentrated to 8–16 mg/ml and subjected to crystallization trials.

Prior to crystallization the purified MtTMEM175-Mb$_{51H01}$ complex was mixed with *E. coli* polar lipids (Avanti) and with *n*-decyl-β-d-maltopyranoside (DM, Anatrace) to a final concentration of 100 μg/ml and 0.3% respectively. Best crystals were obtained in a condition composed of 100 mM Tris-HCl pH 8.5, 150 mM NaCl, 150 mM MgCl$_2$ and 28–30% PEG400 grown at 20°C. After 14 days, the crystals were dehydrated for 3–4 hr using mother liquor with 36% PEG400, cryo-protected and flash-frozen in liquid propane or liquid nitrogen with similar results. The crystals giving the best datasets were additionally soaked in a cryo-protecting solution containing 5 mM KPtCl$_4$ followed by back-soaking in the cryo-protecting solution to get rid of excess platinum. For soaking in cesium and rubidium, 150 mM KCl in the cryo-protecting solution was replaced by 150 mM CsCl and 150 mM RbCl respectively. For the anomalous signal of zinc, crystals of MtTMEM175 WT protein were soaked for 15 min in a cryo-protecting solution containing 0.5 mM ZnSO$_4$ while the T38A mutant was soaked in 0.5–2.5 mM ZnSO$_4$. The mutant MtTMEM175 with a T38A substitution was crystallized in the same condition and crystals were flash frozen in liquid nitrogen.

## Data collection and structure determination

X-ray diffraction data was collected on the X06SA beamline at the Swiss Light Source (SLS) of the Paul Scherrer Institute (PSI) equipped with an EIGER 16M detector (Dectris) at 100K. Data reduction was performed using XDS (*Kabsch, 2010a*) and XSCALE (*Kabsch, 2010b*). The resolution cut off was determined by CC$_{1/2}$ criterion (*Karplus and Diederichs, 2012*). Crystals of MtTMEM175 in complex with Mb$_{51H01}$ belong to space group P4$_2$2$_1$2 (a = 131.2 Å, b = 131.2 Å, c = 132.6 Å), with a solvent content of 64%. Best diffracting crystals of MtTMEM175 WT were obtained after soaking in KPtCl$_4$, but no anomalous platinum signal was detected. For the native data set seven datasets from a single crystal were merged together. Phases were obtained by molecular replacement in PHASER (*McCoy et al., 2007*), using the individual atomic coordinates of MBP (PDB ID: 1ANF) (*Quiocho et al., 1997*), and the nanobody Nb60 (PDB ID: 5JQH) (*Staus et al., 2016*). An initial round of model refinement was performed using REFMAC5 (CCP4 program suite) (*Murshudov et al., 2011*; *Winn et al., 2011*), followed by density modification with Parrot

(*Cowtan, 2010*) and automated model building by Buccaneer (*Cowtan, 2006*). The initial model was improved by iterative cycles of manual model building in Coot (*Emsley et al., 2010*) and refined in Buster-TNT (*Blanc et al., 2004*), yielding excellent geometry (Ramachandran favored/outliers: =95.9%/0.0%) and $R_{work}/R_{free}$ values of 0.209/0.253 (*Supplementary files 1* and *2*). Potassium ion positions were verified by the anomalous signal at high wavelengths ($\lambda = 2.02460$ Å). Refinements using Buster-TNT indicated a high occupancy for $K^+$ at the position of $1K^+$ and lower occupancy for $K^+$ at $2K^+$. Thus, the presence of both, $K^+$ and $Na^+$, at $2K^+$ is possible. Native crystals were soaked with cesium, rubidium, and zinc and the respective ion position determined by the anomalous signal. The anomalous signal for cesium and rubidium ions was strong and identified their positions at the extracellular ion channel entrance (at $1K^+$). The anomalous signal for the data measured at the zinc K-edge ($\lambda = 1.24610$ Å) was weak, suggesting only partial occupancy. The MtTMEM175 model and structure factors (code 6HD8, 6HD9, 6HDA, 6HDB, 6HDC, 6SWR) have been deposited in the Protein Data Bank.

Regions not defined in the electron density include residues 1–3, 283–301 and 484–486 for the $Mb_{51H01}$ expression construct, and residues 1–8 and 241–247 for MtTMEM175 (5.4% in total). Residues 1–3 in the Nb correspond to the N-terminal remainder after 3C cleavage and would be GPS, and Residues 283–301 correspond to residues 166–184 in MBP (numbering without signal peptide) and residues 484–486 correlates to the end of MBP.

The program HOLE (*Smart et al., 1993*) was used to analyze the pore radius in the MtTMEM175 ion conduction pathway and the electrostatic potentials were calculated with the program APBS (*Baker et al., 2001*) with a grid spacing of 0.5 in a range of −5 to +5 kTe. Figure preparation was carried out in PyMOL (Schrödinger LLC). Maps were exported from Coot for use in PyMOL.

## Projection of sequence conservation on the MtTMEM175 structure

Fifteen bacterial TMEM175 sequences were aligned: Nine bacterial sequences obtained from a BLAST search using the sequence of hTMEM175, and five randomly chosen bacterial TMEM175 sequences were aligned with MtTMEM175. The conservation index from this multiple sequence alignment was calculated using AL2CO (*Pei and Grishin, 2001*) and was then used to replace the values for the B-factors in the PDB file of MtTMEM175. Missing parts between the different sequences were assigned a value of −1 by default. More negative values as from the AL2CO conservation index output were set to −1. Conservation index was visualized in the MtTMEM175 structure using cyan-white-magenta colors and with the minimum set to −1 (least conservation, cyan) and the maximum set to 2.8 (maximal conservation, magenta). Sequences used for the alignment were: *Marivirga tractuosa, Lactobacillus rossiae, Mycobacterium sp., Humibacillus sp., Micromonospora chaiyaphumensis, Oscillatoria sp., Azospirillum brasilense, Niastella vici, Streptomyces collinus, Chryseobacterium sp., Streptacidiphilus carbonis, Fulvivirga imtechensis, Methylobacterium extorquens, Deinococcus geothermalis, Paenibacillus curdlanolyticus.*

## TIRF microscopy

HEK293 cells transiently expressing vYFP-tagged TMEM175 proteins were grown on cover slips and decapitated by cold distilled water as described previously (*Biel et al., 2016*). The remaining isolated plasma membrane patches on the glass cover slips were imaged on a Nikon Ti-E microscope (Nikon, Konan, Minato-ku, Tokyo, Japan) with a CFI Apo TIRF 100x objective (NA 1.49, WD 0.12 mm). For TIRF imaging the focus in the back focal plane was moved off-center by controlling the position of a mirror with a single-axis stage M-126. DG controlled by a C-863 Mercury Servo Controller (Physik Instrumente (PI), Karlsruhe, Germany). Plasma membrane patches and potential contamination of remaining cortical ER were stained with red fluorescent CellMask Deep Red (CMDR) and ER-tracker Blue-white DPX (both from Thermo Fisher) respectively. The fluorescent markers were excited/detected as follows: vYFP (488 nm/ 500–550 nm), ER-Tracker (561 nm/ 577.5–646.5 nm), CMDR (647 nm/ 662.5–799.5 nm).

## Cell surface labeling of MtTMEM175 in HEK293 cells using fluorescent Nb$_{51H01}$

Low passage HEK293 cells were grown in µ-Slides VI 0.4 (Ibidi) and transiently transfected with plasmid encoding MtTMEM175 or mock transfected using Fugene following the manufacturer's

protocol. 40 hr after transfection, the cells were washed twice with phosphate buffered saline (PBS) containing 10% fetal bovine serum (FBS) (PBS/FBS) before they were incubated with purified $Nb_{51H01}$-vYFP in PBS/FBS at a concentration of 10 µg/ ml for 20 min. Unbound $Nb_{51H01}$-vYFP was removed by two wash steps (PBS/FBS) before imaging using a Nikon Eclipse Ti2 epifluorescence microscope and a 40x Plan Fluor objective (Nikon) with an $iXon^{EM}$+ 885 EMCCD camera (Andor). For the pre-absorption experiment of $Nb_{51H01}$-vYFP with MtTMEM175, a 4-fold molar excess of MtTMEM175 (from *E. coli*) was incubated with $Nb_{51H01}$-vYFP for 20 min and then applied to the cells as described for the treatment with $Nb_{51H01}$-vYFP alone (DDM concentration in the wells was below the cmc due to dilution).

## Acknowledgements

We are grateful to R Dutzler (University of Zürich) for support, providing materials and lab infrastructure. We thank B Blattmann and C Stutz-Ducommun (Crystallization facility at University of Zürich) and the Swiss Light Source (PSI-Villigen) for excellent support. S Stefanic and P Deplazes (Nanobody Facility, University of Zürich) are acknowledged for immunization of alpacas. Calculations were performed at sciCORE (http://scicore.unibas.ch/) scientific computing center (University of Basel). We thank B Dreier (University of Zürich) and T Sharpe (University of Basel) for help with the MALLS experiment. We thank members of the Dutzler lab/Zürich and Maier lab/Basel for many fruitful discussions and M Steinmetz and his group (PSI-Villigen) for support. AM and GT thank Henry Colecraft (Columbia University) for hospitality in his laboratory. We thank Robert Lehn (TU Darmstadt) for the help with TIRF microscopy. We thank Ronnie Willaert (VUB Brussels) and Charlotte Yvanoff (VUB Brussels) for help with the Nikon Eclipse Ti2 microscope. This work was supported by 2016 Schaefer Research Scholars Program of Columbia University to AM, and European Research Council (ERC) 2015 Advanced Grant 495 (AdG) n. 695078 noMAGIC to AM and GT.

## Additional information

### Funding

| Funder | Grant reference number | Author |
| --- | --- | --- |
| H2020 European Research Council | Advanced Grant 495 (AdG) n. 695078 noMAGIC | Anna Moroni Gerhard Thiel |
| Columbia University | 2016 Schaefer Research Scholars Program | Anna Moroni |

The funders had no role in study design, data collection and interpretation, or the decision to submit the work for publication.

### Author contributions

Janine D Brunner, Conceptualization, Data curation, Formal analysis, Validation, Investigation, Visualization, Methodology, Writing - original draft, Project administration, Writing - review and editing, initiated and designed the project, cloned genes, purified and crystallized proteins, collected data at the synchrotron, analyzed data, generated the phage library, analyzed crystallographic data and carried out refinement and prepared final crystallographic models; Roman P Jakob, Data curation, Formal analysis, Validation, Investigation, Writing - review and editing, analyzed crystallographic data and carried out refinement, provided input for manuscript editing; Tobias Schulze, Data curation, Formal analysis, Validation, Investigation, Methodology, performed and analyzed electrophysiological experiments; Yvonne Neldner, Formal analysis, Investigation, generated the phage library; Anna Moroni, Resources, Data curation, Formal analysis, Funding acquisition, Validation, Investigation, Writing - review and editing, performed and analyzed electrophysiological experiments, provided input for manuscript editing; Gerhard Thiel, Resources, Data curation, Formal analysis, Supervision, Funding acquisition, Validation, Investigation, Visualization, Methodology, Writing - review and editing, performed and analyzed electrophysiological experiments, contributed to manuscript writing; Timm Maier, Resources, Data curation, Formal analysis, Validation, Writing - review and editing, solved the initial crystal structure, analyzed crystallographic data and carried out refinement,

provided input for manuscript editing; Stephan Schenck, Conceptualization, Data curation, Formal analysis, Validation, Investigation, Visualization, Methodology, Writing - original draft, Project administration, Writing - review and editing, initiated and designed the project, cloned genes, purified and crystallized proteins, collected data at the synchrotron and analyzed data, designed the Nb-MBP fusion protein

**Author ORCIDs**
Janine D Brunner (iD) https://orcid.org/0000-0003-4237-9322
Anna Moroni (iD) https://orcid.org/0000-0002-1860-406X
Timm Maier (iD) https://orcid.org/0000-0002-7459-1363

**Decision letter and Author response**
Decision letter https://doi.org/10.7554/eLife.53683.sa1
Author response https://doi.org/10.7554/eLife.53683.sa2

# Additional files

### Supplementary files

• Supplementary file 1. Crystallographic data collection and refinement statistics. The resolution cut-off was determined by $CC_{1/2}$ criterion. For the native data set seven datasets from a single crystal were merged together. For the $K^+/S$ data set, two sets from a single crystal have been merged together. For the $Cs^+$ dataset, three datasets from three crystals have been merged together. Six data sets from two crystals have been merged together for the $Rb^+$ data set. For the $Zn^{2+}$ data set, two data sets from two crystals have been merged together.

• Supplementary file 2. Crystallographic data collection and refinement statistics. The resolution cut-off was determined by $CC_{1/2}$ criterion. For the datasets of the T38A mutant and the T38A mutant soaked with $Zn^{2+}$, four datasets from two crystals have been merged together each.

• Transparent reporting form

### Data availability

Atomic coordinates have been deposited at the Protein Data Bank with the following unique identifiers: 6HD8, 6HD9, 6HDA, 6HDB, 6HDC, 6SWR.

The following datasets were generated:

| Author(s) | Year | Dataset title | Dataset URL | Database and Identifier |
|---|---|---|---|---|
| Brunner JD, Jakob RP, Schulze T, Neldner Y, Moroni A, Thiel G, Maier T, Schenck S | 2019 | Crystal structure of the potassium channel MtTMEM175 in complex with a Nanobody-MBP fusion protein | https://www.rcsb.org/structure/6HD8 | RCSB Protein Data Bank, 6HD8 |
| Brunner JD, Jakob RP, Schulze T, Neldner Y, Moroni A, Thiel G, Maier T, Schenck S | 2019 | Crystal structure of the potassium channel MtTMEM175 with rubidium | https://www.rcsb.org/structure/6HD9 | RCSB Protein Data Bank, 6HD9 |
| Brunner JD, Jakob RP, Schulze T, Neldner Y, Moroni A, Thiel G, Maier T, Schenck S | 2019 | Crystal structure of the potassium channel MtTMEM175 with cesium | https://www.rcsb.org/structure/6HDA | RCSB Protein Data Bank, 6HDA |
| Brunner JD, Jakob RP, Schulze T, Neldner Y, Moroni A, Thiel G, Maier T, Schenck S | 2019 | Crystal structure of the potassium channel MtTMEM175 with zinc | https://www.rcsb.org/structure/6HDB | RCSB Protein Data Bank, 6HDB |
| Brunner JD, Jakob | 2019 | Crystal structure of the potassium | https://www.rcsb.org/ | RCSB Protein Data |

| Author(s) | Year | Dataset title | Dataset URL | Database and Identifier |
|---|---|---|---|---|
| RP, Schulze T, Neldner Y, Moroni A, Thiel G, Maier T, Schenck S | | channel MtTMEM175 T38A variant in complex with a Nanobody-MBP fusion protein | structure/6HDC | Bank, 6HDC |
| Brunner JD, Jakob RP, Schulze T, Neldner Y, Moroni A, Thiel G, Maier T, Schenck S | 2020 | Atomic coordinates | https://www.rcsb.org/structure/6SWR | RCSB Protein Data Bank, 6SWR |

The following previously published dataset was used:

| Author(s) | Year | Dataset title | Dataset URL | Database and Identifier |
|---|---|---|---|---|
| Lee C, Guo J, Jiang Y | 2017 | Crystal structure of a lysosomal potassium-selective channel TMEM175 homolog from Chamaesiphon Minutus | https://www.rcsb.org/structure/5VRE | RCSB Protein Data Bank, 5VRE |

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
