## [Decision Letter]

**Acceptance summary:**

This study describes the structural and functional characterization of a bacterial homolog of the TMEM175 family termed MtTMEM175. This protein family forms endo- and lysosomal K^+^ channels in higher eukaryotes with a protein architecture that is distinct from canonical K^+^ channels. A series of X-ray structures in a presumed non-conducting conformation provide detailed insight into structural features of this protein and into its interaction with cations. Compared to the previously determined structure of a different prokaryotic TMEM175 homologue (CmTMEM175), the described structure shows striking novel features with respect to architectural details and ion-coordination that are related to its presumed functional properties.

**Decision letter after peer review:**

Thank you for submitting your article "Structural basis for ion selectivity in TMEM175 K + channels" for consideration by *eLife*. Your article has been reviewed by three peer reviewers, including László Csanády as the Reviewing Editors and Reviewer #1, and the evaluation has been overseen by Richard Aldrich as the Senior Editor. The following individual involved in review of your submission has agreed to reveal their identity: Eitan Reuveny (Reviewer #3).

The reviewers have discussed the reviews with one another and the Reviewing Editor has drafted this decision to help you prepare a revised submission.

Summary:

This study describes the structural and functional characterization of a bacterial homologue of the TMEM175 family termed MtTMEM175. This protein family forms endo- and lysosomal K^+^ channels in higher eukaryotes with a protein architecture that is distinct from canonical K^+^ channels. A series of X-ray structures in a presumed non-conducting conformation provide detailed insight into structural features of this protein and into the interaction with cations. Compared to the previously determined structure of a different prokaryotic TMEM175 homologue (CmTMEM175), the described structure shows striking novel features with respect to architectural details and ion-coordination that were related to its presumed functional properties. The authors include electrophysiological experiments on WT and mutant channels to decipher the mechanism of K^+^ selectivity and Zn2+ block in this family of channels, and arrive at a conclusion that differs from the earlier interpretation. They propose that upon pore opening, the pore-forming helices undergo a rotation, exposing a different set of side chains to the permeation pathway compared to the ones seen in the closed-state structures. As a consequence, in an open pore K^+^ ions are coordinated by conserved serine and threonine side chains, instead of having to pass through a narrow hydrophobic "nanotube". The work raises interesting controversy in a field that is just emerging, and suggests that the general principles of K^+^ selectivity established for "canonical" K^+^ channels might also hold for the TMEM175 family.

Essential revisions:

1) The authors should confirm that the electrophysiological recordings on MtTM175 indeed measure currents through that channel.

1.1) Although the whole-cell currents of MtTMEM175A recorded from HEK293 cells appear large compared to mock-transfected cells and a point mutation (T38A) has changed ion selectivity of the channel, it would still be interesting to know whether the authors have ever confirmed expression of the protein on a biochemical level. Prokaryotic membrane proteins frequently do not express in mammalian cells and detection of expression would thus provide an additional confirmation that the recorded data are describing the conduction properties of MtTMEM175. In the Materials and methods the authors have described the construction of a protein construct containing Venus-YFP fused to the C-terminus of the protein. Did the authors use this construct for transfection and did they observe fluorescent protein on the plasma membrane?

1.2) Did the authors record single channel data (shown in Figure 2D) with different pipette solutions and did these data correspond to the selectivity properties of the channel measured in whole-cell configuration? Have the authors quantified the cation over anion permeability ratio of MtTMEM175?

2) Interpretation of Zn2+ block of MtTMEM175 should be solidified.

The evidence that Zn2+ acts as pore-blocker of MtTMEM175 is currently very weak. First, the Zn2+ concentrations used in electrophysiology experiments (5 mM) and in the crystal structure (500 mM) are very high. Second, the proposed Zn2+ binding site is in the middle of the membrane, close to L35 (Figure 6A), yet the block shows no voltage dependence. Third, Zn2+ block is unaffected by the T38A mutation.

3) Regarding selectivity of MtTMEM175, the authors arrive at a mechanistic conclusion that differs completely from the earlier conclusion by the Jiang group, but they do not test the effect on selectivity of mutating the residue (L35) proposed to play a key role in the earlier study. Selectivity of the L35A mutant should be studied and the results included in the interpretation.

4) The X-ray data generally appear of high quality and all structures are well refined. Still, in case of the native data (and several other datasets) the authors have used highly redundant datasets, which makes Rmeas a meaningless quantity to estimate data quality. We suggest that the authors provide plots of I/sigI vs. resolution and an extended documentation of the electron density as figure supplements to better illustrate the quality of their data.

---

## [Author Response]

Essential revisions:1) The authors should confirm that the electrophysiological recordings on MtTM175 indeed measure currents through that channel.1.1) Although the whole-cell currents of MtTMEM175A recorded from HEK293 cells appear large compared to mock-transfected cells and a point mutation (T38A) has changed ion selectivity of the channel, it would still be interesting to know whether the authors have ever confirmed expression of the protein on a biochemical level. Prokaryotic membrane proteins frequently do not express in mammalian cells and detection of expression would thus provide an additional confirmation that the recorded data are describing the conduction properties of MtTMEM175. In the Materials and methods the authors have described the construction of a protein construct containing Venus-YFP fused to the C-terminus of the protein. Did the authors use this construct for transfection and did they observe fluorescent protein on the plasma membrane?

We performed a series of experiments to address this point and included these data in the manuscript. First we expressed C-terminally Venus-YFP tagged MtTMEM175 in HEK cells. This construct has not been used for our measurements. In the latter we co-transfected GFP with untagged MtTMEM175. (Both untagged and YFP-tagged MtTMEM175 versions are expressed from the same vector, a pcDNA 3.1 derivative). YFP-tagged MtTMEM175 showed a wide expression pattern with largely intracellular fluorescence (presumably ER and endosomal compartments) as illustrated in Figure 2—figure supplement 2A. This is not unusual upon overexpression, however the bright fluorescence might overwhelm weaker signals from the plasma membrane. We therefore prepared isolated plasma membrane patches by decapitating the cell bodies by osmotic shock. The isolated membrane patches were imaged with TIRF microscopy in which the plasma membrane was stained with a PM-specific dye. Potential contaminations by ER were stained by an ER tracker dye. The results in Figure 2—figure supplement 2B demonstrate an overlap of PM with YFP-tagged MtTMEM175 and support the idea that this protein indeed also reaches the plasma membrane (it might still be only a minor fraction though). We also detected human TMEM175 on the PM. Overexpression has apparently led to a fraction of this protein in the plasma membrane of this otherwise lysosomal/endosomal channel. In this way the channel became, like in previous studies, measurable in whole-cell recordings. From an experimental point of view this is important information.

In a second experiment we generated a fluorescent version of the Nanobody that we used for co-crystallization (Nb_51H01_). For this we expressed the Nb with a C-terminal Venus-YFP tag followed by an SBP tag for purification and produced it in small scale by cytosolic expression in HEK cells. It was shown earlier by us (Schenck et al., 2017) and other labs that Nbs can be expressed as intrabodies. Nb_51H01_ binds an extracellular epitope on MtTMEM175, as evident from our structure (Figure 1A). It is therefore suitable for labeling cells that express this channel on the cell surface. When we applied purified Nb-vYFP to mock and MtTMEM175 (non-fluorescent) transfected, non-permeabilized and unfixed HEK cells, only the MtTMEM175 expressing cells showed a clear fluorescent rim (Figure 2—figure supplement 3). We could abolish this labeling by pre-absorption with the antigen (bacterially expressed and purified MtTMEM175). These data provide further very good evidence that MtTMEM175 is inserted into the plasma membrane of HEK cells (we don’t know to which percentage of the total expression) and can thus account for the measured currents.

In a third example we transiently expressed C-terminally SBP-tagged MtTMEM175 in HEK cells and purified it in small scale with streptavidin beads. We obtained sufficient amounts for a coomassie stain that revealed the expected MW. In addition, the purified protein was injected to a Superdex 5/150 column and elution was monitored by Trp-fluorescence. The protein eluted at almost the same volume as bacterially expressed MtTMEM175 indicating assembled tetramers as shown in Figure 2—figure supplement 4 (the minor difference could be due to the presence of a 3C-site, a myc-tag and the SBP tag, whereas the bacterial sample has been 3C-digested already). The SEC chromatogram revealed no significant higher order aggregates and we therefore think that HEK cells are capable of producing correctly folded (also backed by binding of Nb_51H01_-vYFP) bacterial TMEM175 channels (at least MtTMEM175).

We think these data make our work more conclusive, our data from electrophysiological recordings more trustworthy and we believe it was a very reasonable request from the reviewers to ask for experiments to clarify this. Such information was also lacking in the previous two major publications and our data gives confidence that the methodology can be applied in future investigations.

1.2) Did the authors record single channel data (shown in Figure 2D) with different pipette solutions and did these data correspond to the selectivity properties of the channel measured in whole-cell configuration? Have the authors quantified the cation over anion permeability ratio of MtTMEM175?

The observations of single channel activity were too sporadic for a systematic investigation. At this point the detection of channel activity with stochastic fluctuation is important in the context of the general view in the literature that this lysosomal channel is a leak channel and constantly open (Cang et al., 2015). The present data however underscore that TMEM175 proteins are gated to open and close.

We included data on the cation over anion selectivity using methanesulfonate as a large anion vs. chloride. The channel has no apparent permeability for anions: the reversal voltage was not significantly different when the same recordings were done with standard bath solution containing the large anion methanesulfonate (+0.64 ± 3 mV n=18) or in a bath with 150 mM KCl (2.4 ± 4, n=7).

2) Interpretation of Zn2+ block of MtTMEM175 should be solidified.The evidence that Zn2+ acts as pore-blocker of MtTMEM175 is currently very weak. First, the Zn2+ concentrations used in electrophysiology experiments (5 mM) and in the crystal structure (500 mM) are very high. Second, the proposed Zn2+ binding site is in the middle of the membrane, close to L35 (Figure 6A), yet the block shows no voltage dependence. Third, Zn2+ block is unaffected by the T38A mutation.

We agree with the reviewers that the data on the Zn^2+^ block of MtTMEM175 is not clear or not clearly described in the previous version of our manuscript. In principle we state that a true Zn^2+^ block occurs only in hTMEM175/ animal TMEM175 and that the data on MtTMEM175 is at most hinting at the site of the block and that a different effect likely accounts for the inhibition of MtTMEM175 by Zn^2+^.

The block by Zn^2+^ is very weak in bacterial TMEM175 channels (IC50 ~0.5mM). In hTMEM175 Zn^2+^ is very effective (IC50 ~38µM), and in addition this channel is sensitive to 4-AP (IC50~35µM for Zn^2+^). Regarding the difference in selectivity between hTMEM175 and bacterial TMEM175 channels we have shown that the additional hydrophilic layer of Ser45 underlies this difference. We thus thought that this residue might also account for the very different sensitivity towards blockers. As it was shown before by Jiang’s group, 4-AP and Zn^2+^ are equally effective in hTMEM175 if applied from the extracellular or intracellular side. Hence a block in the pore is likely. Indeed, mutating Ser45 to alanine abolished the sensitivity for both blockers. That means that in the Ser45Ala mutant blockers don’t persist in the pore and that a certain number of ligands must be present for Zn^2+^ and 4-AP to become blocking agents. In the Ser45Ala mutant the layer of threonines (which corresponds to Thr38 in MtTMEM175) is still present, however no inhibition is occurring. This suggests that Thr38 in MtTMEM175 could in turn also not underlie the block by Zn^2+^. And mutation of Thr38 to alanine supports this logic, as the block persists. What is making things more complicated and confusing are the data we got from crystallography. Here, in MtTMEM175 Zn^2+^ can be detected in WT channels in close proximity to Thr38, whereas we could not detect ions in the Thr38Ala mutant, even at higher concentrations. How could this be explained, as the Zn^2+^ ion close to the selectivity filter intuitively suggests the site of the block? Encouraged by your comments we suggest the following scenario: The pore is apparently attracting Zn^2+^ ions in MtTMEM175, and it seems that the electrostatic character of the pore is much influenced by Thr38. Mutating Thr38 to Ala obviously changes this electrostatic landscape and Zn^2+^ ions do not bind in the pore any longer. The presence of Zn^2+^ at this location in WT MtTMEM175 does not mean that a block in MtTMEM175 occurs at this location, although it is tempting to think so. Since the Ser45Ala mutant is insensitive towards Zn^2+^ this would actually be an oddity. But Zn^2+^ is blocking, albeit weakly also bacterial TMEM175 channels. From all the data we have we must conclude that this block is not related to what we observe in hTMEM175. We can currently not explain the mechanism of Zn^2+^ block in MtTMEM175. To describe this difference between the structural and the functional data we still prefer to still show our data on Zn^2+^+ and MtTMEM175 but we move the figures to the supplement. We think it is important to provide the structural information on where Zn^2+^ can be located in the closed pore of the protein and that mutating Thr38 is affecting this potential binding site. However, we refrain from speculating on a causal relationship between the structural and the functional data.

Regarding the concentrations, we clarify here that we have used Zn^2+^ at a final concentration of 0.5 mM for soaks in cryo-solution, not 500 mM. For the Thr38Ala mutant of MtTMEM175 we tested concentrations of 0.5-2.5 mM. We agree that 5 mM Zn^2+^ is a rather high concentration for electrophysiological experiments, but we did not see adverse effects in the experiments with hTMEM175 WT/ Ser45Ala, hence we have no reason to doubt our data regarding hTMEM175.

Another unusual observation is the lack of voltage-dependence for the block by Zn^2+^. We were initially also very surprised by this finding and we can only try to explain this effect from the existing data. It is obvious that the mechanism of Zn^2+^ block is different from the canonical pore block of K^+^ channel by for example Ba^2+^. The location of the Zn^2+^ ion in the pore could help to explain the oddity of a lack of voltage-dependence of this block. MtTMEM175 crystallized in the closed state, yet in crystal soaks the small Zn^2+^ ion could penetrate rather deep into the pore. This means that Zn^2+^ is not pulled as a blocker into the open pore but it can localize to the middle of the membrane plane in the closed state by diffusion. Regarding the electrophysiological data this means that also all closed channels (we currently don’t know the fraction of open/closed channels) could have Zn^2+^ ions in their pores in vicinity to the selectivity filter. From all that we know so far is that the channels fluctuate between open and closed states. Hence the Zn^2+^ ion could therefore be attracted by the selectivity filter before any voltage is applied. Now, when voltage is applied during electrophysiological measurements, the Zn^2+^ ion would be already very near or at the site where the block takes place in vertebrate TMEM175 and thus the voltage dependence of a pore-block, that is a physical consequence of its location in the middle of the membrane plane, would be masked. We agree that this phenomenon needs more investigation to be coherently explained.

We have put now more focus on the human TMEM175 and its sensitivity towards Zn^2+^ and 4-AP for which we have conclusive data. Regarding MtTMEM175 and Zn^2+^ block, we wanted however to make our data available to readers, that is why we moved the data to the supplement (Figure 6—figure supplement 1). We elaborated the discrepancies between MtTMEM175 and hTMEM175 now more precisely in the main text.

3) Regarding selectivity of MtTMEM175, the authors arrive at a mechanistic conclusion that differs completely from the earlier conclusion by the Jiang group, but they do not test the effect on selectivity of mutating the residue (L35) proposed to play a key role in the earlier study. Selectivity of the L35A mutant should be studied and the results included in the interpretation.

In the previous version of the manuscript we only discuss potential events that could explain the loss of selectivity by mutating the presumed gates in TMEM175 channels (Ile in CmTMEM175 and hTMEM175 and Leu in MtTMEM175) such as pore-wetting by comparison to a number of reported cases for similar observations in literature. According to your suggestion, we have now introduced the mutation Leu35Ala in MtTMEM175 and measured the shift of the reversal potential after exchanging K^+^ for Na^+^ in the bath. Unlike other labs, we observed only a minor shift meaning that the channel retained most of its selectivity (in contrast to our results regarding Thr38). From these data we are not able to judge if the flux of ions in the Leu35Ala mutant due to a loss of the potential gate is higher than in the WT channel. This would require normalization of plasma membrane copy numbers. However, this finding supports our hypothesis, but we do of course wonder why our measurements differ from previous investigations by other labs.

Potential reasons:

First, there are no comparable electrophysiological data available for a bacterial homologue. CmTMEM175 (Lee et al., 2017) was not analyzed by whole-cell recordings in HEK cells, but with a rather indirect liposomal flux assay. The flux assay is certainly less accurate for an estimation of changes in selectivity. We thus don’t know if selectivity of an Ile23Ala mutant in CmTMEM175 would yield similar results as ours if it was measured by patch clamp electrophysiology in HEK cells. Second, we analyzed a different homologue; the pore diameter of MtTMEM175 in the closed state could be smaller than in CmTMEM175 as already indicated by the comparison of MtTEM175 and CmTMEM175 crystal structures (Figure 6—figure supplement 2c). Hence conduction of ions in closed conformation might be less pronounced in MtTMEM175 than in pores in the closed conformations of other TMEM175 homologues (e.g. human or CmTMEM175). It could also be that the pore is therefore still not fully wetted and ions can’t permeate due to a vapor lock mechanism. Our result also differs from the data on hTMEM175 in the work by Lee et al.. Again, the effect of mutating the gate in MtTMEM175 might not have exactly the same effect as in the human channel where a loss of selectivity was observed. Additionally, in that work, most of the measurements were performed with an Ile –> Asn mutant that not only reduces the size of the gate, but also the chemical character. Due to the hydrophilic character, Asn can be expected to wet the pore substantially thus attracting cations or water and thereby facilitating interactions with polar molecules in the middle of the pore. Further, Asn is geometrically less restricted for interactions with cations because it exhibits more rotamers than other polar amino acids such as Thr or Ser. This flexibility might impact selectivity as numerous coordination geometries could be present that also enable coordination of Na^+^ ions or hydrates thereof as well as of K^+^ ions. This would then superimpose on the naturally intrinsic selectivity of the Thr and Ser/Ala layers or more generally disturb the selectivity arising from these layers. In addition, interactions of Asn46/Asn271 with Ser45 and Thr49/Thr249 are conceivable that wet the pore more than an alanine at this position.

Apart from these thoughts on the background for the different observations our result for Leu35Ala in MtTMEM175 strongly speaks against a role for these bulky hydrophobic residues in selectivity. If the principle of a biological “hydrophobic nanotube” would apply in TMEM175 channels as proposed by Lee et al. we should observe a significant change in selectivity, as Leu35 should fulfill exactly the same role as it was proposed for the corresponding Ile residues in hTMEM175 and CmTEM175. This is however not the case and thus our additional data further contradicts the idea of a “hydrophobic nanotube-selectivity filter” in this family of channels. We have integrated our new results on the Leu35Ala mutant in the manuscript, see also Figure 5b.

4) The X-ray data generally appear of high quality and all structures are well refined. Still, in case of the native data (and several other datasets) the authors have used highly redundant datasets, which makes Rmeas a meaningless quantity to estimate data quality. We suggest that the authors provide plots of I/sigI vs. resolution and an extended documentation of the electron density as figure supplements to better illustrate the quality of their data.

Many thanks for the suggestion. The reviewers are correct as Rmeas is indeed meaningless for high redundant data. For completeness reason we have given standard parameters of data quality in the Supplementary files 1 and 2. As stated in the Materials and methods section, we have used the CC1/2 criterion as resolution cutoff criterion, see also Manuscript (Paragraph: Data collection and structure determination).

“The resolution cut off was determined by CC1/2 criterion.”

Karplus and Diederichs, (2012).

In addition, we provide now plots of I/sigI vs. resolution as a figure supplement (Figure 1—figure supplement 4) illustrating that merging of multiple data sets lead to data with a high S/N ratio. This has in particular helped in the identification of weak anomalous signal for the low-resolution shells in certain data sets. The plots show that I/sigI is at 1.52 at the resolution cut-off at 2.4 Å for the native data set.

Further, we include now a systematic documentation of the 2Fo-Fc map (native) for all six transmembrane helices together with the model that illustrate the high quality of the data (Figure 1—figure supplements 2-3).